# Prompt Optimization with EASE?
# Efficient Ordering-aware Automated Selection of Exemplars

Zhaoxuan Wu [* 1 2]   Xiaoqiang Lin [* 3]   Zhongxiang Dai [4]   Wenyang Hu [1 3]   Yao Shu [5]   See-Kiong Ng [1 3]
Patrick Jaillet [4]   Bryan Kian Hsiang Low [3]

## Abstract

Large language models (LLMs) have shown impressive capabilities in real-world applications. The capability of *in-context learning* (ICL) allows us to adapt an LLM to downstream tasks by including input-label exemplars in the prompt without model fine-tuning. However, the quality of these exemplars in the prompt greatly impacts performance, highlighting the need for an effective automated exemplar selection method. Recent studies have explored retrieval-based approaches to select exemplars tailored to individual test queries, which can be undesirable due to extra test-time computation and an increased risk of data exposure. Moreover, existing methods fail to adequately account for the impact of exemplar ordering on the performance. On the other hand, the impact of the *instruction*, another essential component in the prompt given to the LLM, is often overlooked in existing exemplar selection methods. To address these challenges, we propose a novel method named EASE, which leverages the hidden embedding from a pre-trained language model to represent ordered sets of exemplars and uses a neural bandit algorithm to optimize the sets of exemplars *while accounting for exemplar ordering*. Our EASE can efficiently find an ordered set of exemplars that *performs well for all test queries* from a given task, thereby eliminating test-time computation. Importantly, EASE can be readily extended to *jointly optimize both the exemplars and the instruction*. Through extensive empirical evaluations (including novel tasks), we demonstrate the superiority of EASE over existing methods, and reveal practical insights about the impact of exemplar selection on ICL, which may be of independent interest.

## 1. Introduction

Large language models (LLMs) have recently drawn significant attention and have been widely deployed in various real-world scenarios (Maslej et al., 2024; Minaee et al., 2024; Zhao et al., 2023) due to their strong capabilities. Of note, a particularly impressive capability of LLMs is *in-context learning* (ICL): LLMs can learn from a handful of input-label demonstrations (i.e., *data exemplars*) included in its prompt to perform a downstream task (Brown et al., 2020; Liu et al., 2022). ICL allows us to adapt an LLM to a downstream task without fine-tuning the model parameters (Li et al., 2023; Mosbach et al., 2023). However, the ICL performance is heavily dependent on the data exemplars in the prompt (Albalak et al., 2024; Liu et al., 2022; Rubin et al., 2022). Therefore, to maximize the ICL performance, it is of paramount importance to carefully select the set of exemplars. Unfortunately, exemplar selection for ICL is challenging as the mechanism of ICL is complicated and unknown, and this difficulty is further aggravated for black-box LLMs with only API access (e.g., GPT-4 (OpenAI et al., 2024), Gemini Ultra (Team et al., 2024)).

A large number of existing works on exemplar selection for ICL have considered *retrieval-based* methods (Albalak et al., 2024; Liu et al., 2022; Rubin et al., 2022). Specifically, they aim to develop a retriever model such that for every test query, the retriever model retrieves the corresponding best set of exemplars tailored to this particular query (Albalak et al., 2024; Ye et al., 2023). Therefore, at test time, the retriever needs to be deployed for *every test query*, which could incur significant additional computational overheads especially when the number of test queries is large (Luo et al., 2024). This is likely to hinder their ease of adoption in practice because of the prolonged response time and the additional complexity from running the retriever for every query. Moreover, another drawback of these retrieval-

---
[*]Equal contribution  [1]Institute of Data Science, National University of Singapore  [2]Integrative Sciences and Engineering Programme, National University of Singapore  [3]Department of Computer Science, National University of Singapore  [4]LIDS and EECS, Massachusetts Institute of Technology  [5]Guangdong Lab of AI and Digital Economy (SZ). Correspondence to: Zhongxiang Dai <daizx@mit.edu>.

*Proceedings of the 1st Workshop on In-Context Learning at the 41st International Conference on Machine Learning*, Vienna, Austria. 2024. Copyright 2024 by the author(s).

based methods is that they may lead to increased privacy risks. Compared to using a fixed set of exemplars for all test queries submitted to the LLM provider (e.g., via an API), using a different set of exemplars for every test query may lead to greater data exposure, i.e., more data exemplars being exposed to the LLM provider. Privacy issues have become increasingly prominent in discussions surrounding LLMs (Neel & Chang, 2024), with evidence of user data being output by these models (White, 2023). Therefore, retrieval-based methods are undesirable in scenarios where such privacy concerns are important. Given these two shortcomings of retrieval-based methods, it is imperative to develop exemplar selection methods that generalize to test queries and are capable of choosing a fixed set of exemplars that perform well for all test queries for a task.

Another major limitation of existing exemplar selection methods (including retrieval-based and other methods) is their inability to account for the impact of the *ordering of the exemplars* within a subset (Lu et al., 2022; Perez et al., 2021; Zhao et al., 2021). Specifically, to select a subset of $k$ exemplars, previous works have either relied on simple heuristics (e.g., selecting the top-$k$ exemplars based on the score from a retriever (Luo et al., 2024)) or used subset selection techniques which are able to account for the inter-relationships among the exemplars within a subset (Chang & Jia, 2023; Gupta et al., 2023; Levy et al., 2023; Li & Qiu, 2023; Ye et al., 2023). Due to the computational complexity caused by the combinatorial search space, these works based on subset selection have often employed an iterative greedy strategy to sequentially select the subset of exemplars. It is also non-trivial to extend these existing works to consider exemplar ordering. However, it has been repeatedly verified that the ordering of the exemplars has a significant impact on the performance of ICL (Lu et al., 2022; Perez et al., 2021; Zhao et al., 2021). To the best of our knowledge, the impact of the exemplar ordering during the selection process remains inadequately addressed by these previous works that relied on simple heuristics (e.g., random ordering) and approximations (e.g., greedy strategy).

In addition, when optimizing the set of exemplars to improve the performance of the LLM, existing methods have often ignored the impact of the *instruction*, which is another essential component in the prompt given to the LLM (Lin et al., 2023). Specifically, previous works on exemplar selection often use a fixed pre-determined instruction or do not include any instruction at all in the prompt (Chang & Jia, 2023; Li & Qiu, 2023; Liu et al., 2022; Rubin et al., 2022). Meanwhile, existing works on instruction optimization for LLMs typically include a fixed manually selected set of exemplars in the prompt (Fernando et al., 2023; Hu et al., 2024; Lin et al., 2023; Yang et al., 2024) or simply adopt the zero-shot setting (i.e., without using any exemplar) (Chen et al., 2023a; Guo et al., 2024). However, these

two lines of work have separately demonstrated that both the exemplars and the instruction have significant impacts on the performance of the LLM. Therefore, the existing works that optimize these two components separately are unable to account for their interactions, which may be crucial for further boosting the performance of LLMs. This leaves considerable untapped potential in enhancing the performance of the LLM through ICL, which can be achieved by *jointly optimizing the instruction and the exemplars*.

In this work, we propose a novel exemplar selection algorithm that addresses the above-mentioned challenges faced by existing works (with detailed discussions of related work in App. A). We formulate exemplar selection as a black-box optimization problem, in which every input in the domain corresponds to a sequence of $k$ exemplars and its output represents the ICL performance achieved by including this sequence in the prompt for the LLM. Given this formulation, for every sequence of $k$ exemplars in the domain, we adopt the embedding from a powerful pre-trained language model as its continuous representation, and train a neural network (NN) to predict its ICL performance. Based on the trained NN, we use a neural bandit algorithm to find the optimal exemplar sequence in a query-efficient manner. Our Efficient ordering-aware Automated Selection of Exemplars (EASE) algorithm offers several significant benefits:

- Our EASE can *find an efficacious sequence of $k$ exemplars* (which performs well for all test queries from a task) in a *query-efficient manner* (i.e., it only needs to test a small number of exemplar sequences). This can mainly be attributed to our algorithmic design, which allows us to use neural bandits to balance the *exploration* of the space of exemplar sequences and the *exploitation* of the predicted performance from the NN in a principled way. Importantly, in contrast to retrieval-based methods, our EASE *requires no test-time computation* to select test query-specific exemplars.

- Our EASE naturally *takes into account the ordering of the exemplars* when maximizing the ICL performance. This is because, for the same subset of exemplars, a different ordering leads to a different pre-trained embedding, which allows the trained NN (that takes the embedding as input) to predict the performances of different orderings of these exemplars. We have made our EASE computationally feasible for large search spaces of exemplar sequences using a technique based on optimal transport (OT), which reduces the computational cost of EASE while preserving its strong performance by imposing an implicit preference towards exemplars that are more relevant to the task (Sec. 3.2).

- Our EASE is readily extended to *jointly optimize the exemplars and instruction* in the prompt. This is achieved by augmenting the domain of exemplar sequences with the instruction (Sec. 3.3).

These advantages of our EASE algorithm allow it to significantly boost the performance of ICL. To empirically validate this, we compare our EASE algorithm with a comprehensive suite of baselines in a variety of tasks. To begin with, we show that our EASE consistently outperforms previous baselines in large number of benchmark tasks (Sec. 4.1). Next, by using our EASE algorithm in a novel experiment, we unveil an interesting insight about ICL: The selection of exemplars is more important when the LLM has less knowledge about the task (Sec. 4.2). Based on this insight, we design a set of novel experiments in which the selection of exemplars has an important impact on the ICL performance, and use these experiments to further demonstrate the superiority of our EASE (Sec. 4.3). Furthermore, we showcase the ability of our EASE to jointly optimize the exemplars and the instruction to enhance the performance of the LLM even further (Sec. 4.4). We also included a retrieval-based extension of EASE to deal with large exemplar set sizes (Sec. 4.5). Of note, our novel experimental designs, as well as the insights from them, *may be of independent interest for future works on ICL and beyond.*

## 2. Problem setting

We are given a set of data exemplars $D = \{e_i = (x_i, y_i)\}_{i=1}^n$ of size $n$, where $x_i$ and $y_i$ correspond to the input and output text, respectively. The data exemplars describe a downstream task, and we aim to select $k$ of them to form the optimal in-context exemplars sequence $E$ for accurate output generation when prompting a black-box LLM $f(\cdot)$. Due to the sequential nature of the natural language input, the exemplar sequence is ordered such that $E = (e_1, e_2, \ldots, e_k)$ where $e_i$ denotes the $i$-th exemplar chosen. For every input $x$, we prepend it with a sequence of exemplars $E$ to generate a response $\hat{y}$ from the black-box LLM (e.g., through calling the API), following

$$\hat{y} = f([\underbrace{e_1, e_2, \ldots, e_k}_{\text{context}}, x]) = f([E, x]) .$$

Here, the number of exemplar $k$ in the context can be determined by the future budget of inference in practice. A larger $k$ corresponds to a longer sequence of context tokens to be prepended to the test input $x$ during inference, which leads to higher query costs (e.g., associated with the API calls). Alternatively, it is common to decide $k$ depending on the context window size of the target LLM $f(\cdot)$.

Since the model architecture and the internal working of the black-box LLM $f(\cdot)$ is unattainable, we formulate exemplar selection as a black-box optimization problem over the space of permutations $\Omega \triangleq \{E : |E| = k\}$ of size $n^k$ (i.e., having $n$ choices for each of the $k$ positions in the sequence),

$$\max_{E \in \Omega} F(E) \triangleq \mathbb{E}_{(x,y) \in D_V} [s(f(E, x), y)] \qquad (1)$$

where $s(\cdot, \cdot)$ is a score function for the output against the ground truth, $D_V$ is the held-out validation set and $|E| = k$ denotes that the number of exemplars in $E$ is $k$. Therefore, the exemplar sequence $E$ found represents a fixed ordered set of exemplars that apply to every data point in the validation set. Note that our formulation considers exemplars jointly as *ordered* text in the sequence $E$. This space of permutation $\Omega$ is also a lot larger than the (unordered) combinatorial search space considered in the subset selection formulation of exemplar selections in (Chang & Jia, 2023; Gupta et al., 2023; Li & Qiu, 2023; Ye et al., 2023). We show the superior performance of our method against subset selection methods in our experiments (Sec. 4).

## 3. Automated selection of exemplars

### 3.1. NeuralUCB for query-efficient optimization of exemplar sequences

We propose to use neural bandits to iteratively maximize the black-box objective function (1). At each iteration $t$, the neural bandits algorithm selects the next input query based on the belief of the objective given all past $t - 1$ observations $O_{t-1} \triangleq \{(E_i, s_V(E_i))\}_{i=1}^{t-1}$ where $E_i$ and $s_V(E_i)$ are the exemplar sequence and corresponding validation score at iteration $i$, respectively. Here, the validation score is a realization of the objective function (1), so $s_V(E) = 1/|D_V| \sum_{(x,y) \in D_V} s(f(E, x), y)$.

Inspired by Lin et al. (2023) who have shown impressive performances of the neural upper confidence bound (NeuralUCB) acquisition function (Zhou et al., 2020) on instruction optimization, we adopt the NeuralUCB approach to our novel black-box exemplar selection formulation in (1). We propose to use a neural network (NN) to directly learn the mapping from a general-purpose hidden embedding of input exemplar sequences to the validation score. Specifically, we propose to use a embedding model $h(\cdot)$ and train a network $m(h(E); \theta_t)$ at iteration $t$ such that

$$\theta_t = \arg\min_\theta \mathbb{E}_{(E, s_V(E)) \in O_{t-1}} \ell \left( m(h(E); \theta), s_V(E) \right) \tag{2}$$

where $\ell$ is the mean squared error (MSE) loss. The embedding model is pre-trained on large text corpora and provides a powerful latent representation of the input. For example, pre-trained models like MPNet (Song et al., 2020) and black-box APIs like OpenAI text embedding are commonly used for clustering, semantic search and classification. Importantly, even for the same subset of exemplars, a different ordering leads to different embedding, hence $h(E)$ captures both the content and the ordering of the exemplars in $E$.

Then, the trained NN can be used to iteratively select the next exemplar sequence $E_t$ to query:

$$E_t = \arg\max_{E \in \Omega} \text{NeuralUCB}_t(E),$$
$$\text{NeuralUCB}_t(E) \triangleq m(h(E); \theta_t) + \nu_t \sigma_{t-1}(h(E); \theta_t), \tag{3}$$

where $m(h(E); \theta_t)$ is the predicted score, $\sigma_{t-1}(h(E); \theta_t)$ is the NN's uncertainty about the score of $h(E)$ and $\nu_t$ is a hyperparameter that balances the two terms. Using NeuralUCB, our EASE balances the *exploration* of the space of exemplar sequences $E$ and the *exploitation* of the predicted score from the NN in a principled way.

This application of NeuralUCB in our work differs from that of Lin et al. (2023) in three main aspects. Firstly, the input to our NN is the embedding of *exemplar sequences* rather than the latent representation of the *instructions* from (Lin et al., 2023). Secondly, we relax the requirement of a separate white-box LLM from (Lin et al., 2023) and only need black-box access to the embedding model $h(\cdot)$. Thirdly, our application to exemplar selection dramatically increases the search space from finite candidate instructions (Lin et al., 2023) to the space of exemplar sequences. Particularly, the explosion of the size of the search space poses significant difficulties to optimization, which we will address next.

### 3.2. Reducing computational cost through optimal transport (OT)

Directly applying NeuralUCB to exemplar selection is challenging due to the enormous search space of all permutations of exemplars on which the acquisition values (3) have to be evaluated. For example, selecting 5 exemplars out of 100 requires $100^5$ evaluations which is far from being practical. It is natural to search over a *domain space* $Q_t = \{E^{(j,t)}\}_{j=1}^q$ with a smaller number $q$ of exemplar sequences in each iteration $t$. However, uniformly sampling such a reduced space $Q_t$ from $\Omega$ could be sub-optimal, because a small $q$, which is required to make our algorithm computationally feasible, is highly likely to discard important exemplar sequences (we verify this in ablation experiments in Sec. 4.5 and App. D.3). To this end, we make a large $q$ feasible by introducing a novel technique based on optimal transport (OT) to further select from $Q_t$ a subset $Q_t'$ of $q' < q$ *relevant* exemplar sequences. This technique can simultaneously (a) reduce the computational cost (since $q' \ll \Omega$) and (b) preserve our performance (since we can now search over a large domain space $Q_t$) by imposing an implicit preference towards exemplars in $Q_t$ that are more relevant to the task.

Given probability measures $\mu_s$ and $\mu_v$ over space $\mathcal{Z}$, the OT distance between $\mu_s$ and $\mu_v$ is defined as

$$OT(\mu_s, \mu_v) = \min_{\pi \in \Pi(\mu_s, \mu_v)} \int_{\mathcal{Z}^2} c(z, z') d\pi(z, z') \quad (4)$$

where $\Pi(\mu_s, \mu_v) = \{\pi \in \mathcal{P}(\mathcal{Z} \times \mathcal{Z}) | \int_{\mathcal{Z}} \pi(z, z') dz = \mu_s, \int_{\mathcal{Z}} \pi(z, z') dz' = \mu_v\}$ is a collection of couplings between two the distribution $\mu_s$ and $\mu_v$, and $c : \mathcal{Z} \times \mathcal{Z} \to \mathbb{R}^+$ is some symmetric positive cost function. In our problem, we propose to use the space of embedding from $h(\cdot)$ as $\mathcal{Z}$ because the embedding captures the semantics of the

exemplars with a fixed-dimensional vector. As cosine similarity is usually used to train embedding models such as MPNet (Song et al., 2020) and sentence-BERT (Reimers & Gurevych, 2019), we propose to use the following cost function $c(h(e), h(e')) = 1 - sim_{cos}(h(e), h(e'))$ where $sim_{cos}(h(e), h(e'))$ measures the cosine similarity between the embedding of two exemplars $e$ and $e'$. Given a sampled subset $S = \{e_i\}_{i=1}^k$, we define a discrete measure $\mu_s = \frac{1}{k} \sum_{i=1}^k \delta(h(e_i))$ where $\delta$ is the Dirac function. Likewise, we define $\mu_v$ for the validation set $D_V$.

Intuitively, a smaller value of (4) indicates that the subset of exemplars is more similar to the validation set and is hence more *relevant* to the task. This allows us to select the relevant exemplar sequences from $Q_t$ to form the smaller subset $Q_t'$, on which the acquisition values (3) are evaluated. Consequently, we can increase the size $q$ of $Q_t$ by hundreds of folds while being computationally feasible since the most expensive computation (i.e., computing embeddings for all exemplar sequences in $Q_t$) is not needed. Note that the extra computation incurred by OT is minimal since the embedding of every data exemplar in $D$ and $D_V$ can be pre-computed and reused. Therefore, OT helps us examine a large number of permutations among the space of all permutations without significantly increasing the computation, which helps mitigate the problem of the exploding search space.

### 3.3. Natural extension to jointly optimize instructions and exemplars

Our algorithm can further boost the performance of ICL for LLMs by jointly optimizing the exemplars and the instruction. A natural extension of our formulation in Sec. 2 allows the instruction, being another essential component of the LLM prompt, to be simultaneously optimized with exemplars. This ensures the optimal matching between instructions and exemplars to achieve a fully automated pipeline with superior performance. Specifically, our formulation naturally extends to $E = (p, e_1, e_2, \ldots, e_k)$ where $p \in P$ is an instruction from a candidate space/set $P$, i.e., $Q_t' \leftarrow P \times Q_t'$. Subsequently, $p$ can be intuitively treated as another type of "exemplar" from a new space $P$ and optimized in conjuction with the exemplars. In practice, the fixed set $P$ of candidate instructions can be generated by the black-box model through techniques such as APE (Zhou et al., 2023), PromptBreeder (Fernando et al., 2023), etc. This extension is not only simple in its implementation but also proven to be effective in our experiments in Sec. 4.4.

### 3.4. Our EASE algorithm

In iteration $t$ of EASE, we first use the historical observations of the exemplar sequence and score pairs $\{(E_i, s_V(E_i))\}_{i=1}^{t-1}$ to train the score prediction NN. Then, we perform sampling to obtain the domain space $Q_t$, which will be further refined to a set $Q_t'$ of top-$q'$ candidates based

on OT distances. The space of exemplar sequence $E$ can be optionally augmented with instructions $p \in P$ from a set $P$ of instruction candidates. Subsequently, we find the optimal $E$ that maximizes the NeuralUCB acquisition function. The selected exemplar sequence $E$ is then evaluated against the black-box LLM $f(\cdot)$ using the validation dataset $D_V$, obtaining a new observed pair of $(E, s_V(E))$. This process is repeated until the query budget $T$ is exhausted. An overview of the algorithm is presented in Alg. 1.

---

**Algorithm 1:** `EASE`

**Require :** Data exemplars set $D$, validation set $D_V$, length of exemplars $k$, total budget $T$, number of initial rounds $T_{\text{init}}$, sampling size $q'$, black-box target model $f(\cdot)$, embedding model $h(\cdot)$, neural network $m(\cdot; \theta)$, *(optional)* instruction set $P$.

1  Initialize $\{(E_i, s_V(E_i))\}_{i=1}^{T_{\text{init}}}$ with $T_{\text{init}}$ randomly sampled $\{E_i\}_{i=1}^{T_{\text{init}}}$ from $D$;
2  **for** $t = T_{init}$ **to** $T$ **do**
3      Following (2), use observations $\{(E_i, s_V(E_i))\}_{i=1}^{t-1}$ to train the NN $m(\cdot; \theta_t)$ parameter $\theta_t$;
4      Sample sequences $Q_t$ and select top-$q'$ sequences $Q'_t$ with smallest OT distances;
5      *(Optional)* Augment each $E \in Q'_t$ with instructions $p \in P$, i.e., $Q'_t \leftarrow P \times Q'_t$;
6      Following (3), select the next query $E_t = \arg\max_{E \in Q'_t} \text{NeuralUCB}_t(E)$;
7      Evaluate $E_t$ on the black-box model to obtain the validation score $s_V(E_t)$;
8  **return** $E^* = \arg\max_{E_t : t \in \{1,...,T\}} s_V(E_t)$

---

## 4. Experiments

We compare `EASE` with the following comprehensive suite of baselines. They include *subset selection methods* with (a) a determinantal point process (**DPP**) metric adapted from (Ye et al., 2023), and (b) maximum mean discrepancy (**MMD**) (Sejdinovic et al., 2013), (c) optimal transport (**OT**) (Villani, 2021) metrics. We also adapt retrieval-based methods to our setting using a new retrieve-then-sample strategy based on the validation set (details in App. C.2 and Sec. 4.5), specifically with the classical (d) **Cosine** similarity and (e) **BM25** (Robertson & Zaragoza, 2009) retrievers. We also compare with existing exemplar selection baselines using (f) an active selection policy learned using reinforcement learning (**Active**) (Zhang et al., 2022), and (g) an exemplar influence metric (**Inf**) (Nguyen & Wong, 2023). Additionally, we propose two more new strong baselines: (h) **Evo** which mutates exemplars through evolutionary strategies and (i) **Best-of-N** which explores the exemplar space uniformly until the query budget is exhausted. More implementation details are found in App. C.2. The number

of exemplars in the in-context prompt is set to $k = 5$. The black-box query budget is 165 evaluations following Lin et al. (2023). Since the effectiveness of the optimization is directly reflected by the value of the objective function in (1), we report the validation accuracy in the following experiments unless otherwise specified. The test accuracy tables are presented in App. D.11.

### 4.1. Empirical study of performance gains on various benchmark tasks

To study the effectiveness of `EASE`, we conduct empirical comparisons with baseline methods using the Instruction Induction (II) benchmark tasks (Chen et al., 2023a). We test on tasks that contain more than 100 training examples for selection. The results are shown in Tab. 1. Our `EASE` achieves the best performance in 17 out of 19 language tasks. We observe that the subset selection methods such as DPP, MMD and OT are ineffective in practice, implying that choosing subsets alone without considering the order information is inadequate. While the retrieval-based methods Cosine and BM25 have demonstrated success in retrieving test-sample-based exemplars (Rubin et al., 2022), it is not as effective in finding the best exemplars that generalize to the entire task. This is because the exemplars retrieved at the instance level may not work well for the entire validation set. The Active baseline requires training an active exemplar selection policy through the data-intensive reinforcement learning process, which explains its ineffectiveness under budget constraint. The Inf baseline is not efficient in subset sampling and disregards exemplar ordering, leading to worse results. Our proposed Evo and Best-of-N are the most competitive baselines both with best performance in 9 tasks.

We discover that while `EASE` consistently outperforms or matches the baselines across tasks, for some tasks, most baselines achieve good performances. We hypothesize that *the effect of exemplar selection diminishes as the model gains more knowledge about the task*. This explains the instances where the choice of in-context exemplars has minimal impact on the performance, as the black-box model (i.e., GPT-3.5 Turbo) has been pre-trained on these publicly available language tasks. As suggested by Brown et al. (2020), it is highly likely that GPT-3 has been trained on common benchmark datasets potentially contained in Common Crawl due to data contamination. Hence, further providing high-quality in-context exemplars could hardly improve the performance of the model on these well-trained tasks. Another possibility could be that the exemplars mostly serve as the context for adapting to the new formatting rules of the specific task, i.e., the LLM is not utilizing the semantic contents of the exemplars to learn the underlying input-output relations. This aligns with the discoveries of Min et al. (2022) and provides a potential explanation for the surprising behavior of LLMs. Therefore, the II dataset might not be

Table 1: Average accuracy ± standard error achieved by the best exemplar sequence discovered by different algorithms over 3 independent trials. For better distinguishability, we do not include easy tasks here (i.e., with 100% accuracy across baselines) and show full results in Tab. 5 of App. D.1.

| | DPP | MMD | OT | Cosine | BM25 | Active | Inf | Evo | Best-of-N | EASE |
|---|---|---|---|---|---|---|---|---|---|---|
| antonyms | 70.0±0.0 | 80.0±0.0 | 81.7±1.7 | 85.0±0.0 | 85.0±0.0 | 80.0±0.0 | 86.7±1.7 | 88.3±1.7 | **90.0±0.0** | **90.0±0.0** |
| auto_categorization | 3.3±1.7 | 8.3±1.7 | 0.0±0.0 | 25.0±0.0 | 16.7±1.7 | 10.0±2.4 | 21.7±1.7 | 21.7±1.7 | 20.0±0.0 | **30.0±0.0** |
| diff | 0.0±0.0 | 0.0±0.0 | 0.0±0.0 | 0.0±0.0 | 0.0±0.0 | 0.0±0.0 | **100.0±0.0** | **100.0±0.0** | **100.0±0.0** | **100.0±0.0** |
| larger_animal | 70.0±0.0 | 91.7±1.7 | **100.0±0.0** | **100.0±0.0** | **100.0±0.0** | 66.7±1.4 | **100.0±0.0** | **100.0±0.0** | **100.0±0.0** | **100.0±0.0** |
| negation | **95.0±0.0** | **95.0±0.0** | **95.0±0.0** | **95.0±0.0** | **95.0±0.0** | **95.0±0.0** | **95.0±0.0** | **95.0±0.0** | **95.0±0.0** | **95.0±0.0** |
| object_counting | 55.0±2.9 | 56.7±1.7 | 48.3±1.7 | 61.7±1.7 | 66.7±1.7 | 51.7±1.4 | 63.3±4.4 | 70.0±0.0 | 70.0±0.0 | **73.3±1.7** |
| orthography_starts_with | 20.0±2.9 | 35.0±0.0 | 61.7±1.7 | 78.3±1.7 | 70.0±0.0 | 43.3±1.4 | 70.0±2.9 | 75.0±0.0 | 78.3±1.7 | **80.0±0.0** |
| rhymes | 60.0±0.0 | 51.7±1.7 | 0.0±0.0 | **100.0±8.2** | 80.0±0.0 | 65.0±8.2 | 70.0±13.2 | **100.0±0.0** | **100.0±0.0** | **100.0±0.0** |
| second_word_letter | 10.0±2.9 | 30.0±0.0 | 28.3±1.7 | 50.0±0.0 | 50.0±0.0 | 26.7±8.3 | 40.0±0.0 | 46.7±1.7 | 50.0±0.0 | **53.3±1.7** |
| sentence_similarity | 20.0±0.0 | 21.7±3.3 | 40.0±2.9 | 46.7±1.7 | 53.3±1.7 | 5.0±4.1 | 18.3±6.7 | 45.0±0.0 | 51.7±1.7 | **56.7±1.7** |
| sentiment | 85.0±0.0 | 90.0±0.0 | 85.0±0.0 | 96.7±1.7 | **100.0±0.0** | 85.0±4.1 | 91.7±1.7 | **100.0±0.0** | **100.0±0.0** | **100.0±0.0** |
| sum | 0.0±0.0 | 0.0±0.0 | 0.0±0.0 | 0.0±0.0 | 0.0±0.0 | 0.0±0.0 | **100.0±0.0** | **100.0±0.0** | **100.0±0.0** | **100.0±0.0** |
| synonyms | 10.0±0.0 | 25.0±0.0 | 20.0±0.0 | **35.0±0.0** | 30.0±0.0 | 3.3±1.4 | 26.7±1.7 | 30.0±0.0 | 30.0±0.0 | 30.0±0.0 |
| taxonomy_animal | 43.3±4.4 | 40.0±2.9 | 46.7±1.7 | 85.0±2.9 | 80.0±0.0 | 45.0±6.2 | 70.0±5.0 | 80.0±0.0 | 80.0±0.0 | **88.3±1.7** |
| translation_en-de | **90.0±0.0** | 80.0±0.0 | 80.0±0.0 | **90.0±0.0** | 85.0±0.0 | 56.7±13.0 | **90.0±0.0** | **90.0±0.0** | **90.0±0.0** | **90.0±0.0** |
| translation_en-es | 90.0±0.0 | **100.0±0.0** | 96.7±1.7 | **100.0±0.0** | **100.0±0.0** | 96.7±1.4 | 98.3±1.7 | **100.0±0.0** | **100.0±0.0** | **100.0±0.0** |
| translation_en-fr | 76.7±1.7 | 76.7±1.7 | 81.7±1.7 | 85.0±0.0 | 85.0±0.0 | 81.7±1.4 | 85.0±0.0 | 86.7±1.7 | 85.0±0.0 | **88.3±1.7** |
| word_sorting | 26.7±1.7 | 88.3±1.7 | 88.3±1.7 | 90.0±0.0 | 71.7±1.7 | 80.0±0.0 | 88.3±1.7 | **93.3±1.7** | 91.7±1.7 | 90.0±0.0 |
| word_unscrambling | 68.3±1.7 | 56.7±1.7 | 71.7±1.7 | 75.0±0.0 | 76.7±1.7 | 63.3±3.6 | 66.7±1.7 | 75.0±0.0 | 75.0±0.0 | **78.3±1.7** |
| # best-performing tasks | 2 | 2 | 2 | 6 | 4 | 1 | 5 | 9 | 9 | 17 |

the most suitable one to test EASE. Next, we verify the above hypothesis that the effect of exemplar selection diminishes as the model gains more knowledge about the task in Sec. 4.2, and propose more suitable families of datasets for exemplar selection in Sec. 4.3.

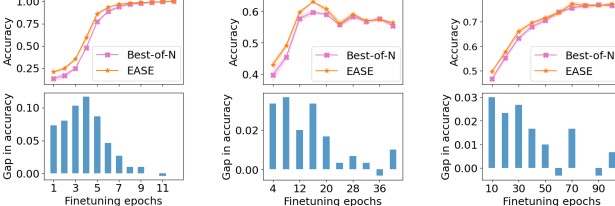

Figure 1: From left to right, the tasks are taxonomy animal, sentence similarity and object counting. The performance gaps between EASE and the Best-of-N baseline diminish as the LLM is finetuned.

## 4.2. Empirical validation of the hypothesis through progressive finetuning

We hypothesize that *the effect of exemplar selection diminishes as the model gains more knowledge about the task*. To gain insights on this hypothesis, we study an open-source white-source Vicuna model instead of the black-box GPT-3.5 model that is only accessible through API. We progressively finetune the while-box language model, and examine whether the extent of finetuning on a specific task diminishes the importance of in-context exemplars when prompted.

Our results are in Fig. 1. Compared to the most competitive baseline, Best-of-N, *the gain from exemplar selection using our EASE diminishes as the model is finetuned on the dataset of the respective tasks for more epochs*. Across the three tasks shown in Fig. 1, using EASE originally has a performance gain of about 3%-10% and this gain slowly diminishes to 0 as finetuning progresses. This verifies our hypothesis and calls for further investigation of the exemplar selection performance of our EASE on tasks that have not been seen by the model, which we conduct in Sec. 4.3.

## 4.3. New families of "out-of-distribution" tasks that emphasize in-context reasoning

We propose three new families of "out-of-distribution" tasks (i.e., loosely referred to as tasks on which the LLM is not already well trained) that highlight the importance of high-quality exemplars, which could also be of independent inter-

est. The new tasks require the LLM to learn the underlying function/relationship in the input prompt in order to perform reasoning during inference, and are hence more sensitive to the quality of the exemplars.

**Rule-based tasks.** Given our insight on the impact of the model's existing knowledge about the task (Sec. 4.2), we propose rule-based tasks that contain novel rules that the LLM has not learned before. A key characteristic of these tasks is that the model has to extract the underlying relationships among the provided in-context exemplars and directly use the relationship for test-time inferences. For example, we construct the linear regression (**LR**) task where the input takes the form demonstrated in Example 2 (see App. C.1). The underlying relationship in this example is $y = ax + b$, with $a = -4$ and $b = 6$ in this specific Example 2. Without further instructions, the model is supposed to rely on the provided in-context exemplars to implicitly infer the regression task, recover the coefficients $a, b$ for linear regression, and then directly apply it to the test sample (e.g., for test input 117, compute $-4 \times 117 + 6 = -462$ and output $-462$). The results are in Tab. 2. For the clean dataset (i.e., 0% noise), EASE outperforms the most competitive baseline by 8.3% in absolute accuracy. Note that EASE has greater ad-

Table 2: Average accuracy $\pm$ standard error over 3 independent trials achieved by different algorithms on the new families of "out-of-distribution" tasks.

| Type | Task | Noise | DPP | MMD | OT | Cosine | BM25 | Active | Inf | Evo | Best-of-N | EASE |
|---|---|---|---|---|---|---|---|---|---|---|---|---|
| Rule-based tasks | LR | 0% | $31.7_{\pm1.7}$ | $38.3_{\pm3.3}$ | $50.0_{\pm0.0}$ | $71.7_{\pm1.7}$ | $70.0_{\pm0.0}$ | $36.7_{\pm1.4}$ | $56.7_{\pm7.3}$ | $61.7_{\pm1.7}$ | $66.7_{\pm1.7}$ | $\mathbf{80.0_{\pm2.9}}$ |
| | | 10% | $8.3_{\pm1.7}$ | $36.7_{\pm1.7}$ | $48.3_{\pm1.7}$ | $61.7_{\pm1.7}$ | $61.7_{\pm1.7}$ | $0.0_{\pm0.0}$ | $58.3_{\pm4.4}$ | $60.0_{\pm0.0}$ | $65.0_{\pm2.9}$ | $\mathbf{73.3_{\pm1.7}}$ |
| | | 30% | $10.0_{\pm0.0}$ | $28.3_{\pm1.7}$ | $46.7_{\pm1.7}$ | $63.3_{\pm1.7}$ | $60.0_{\pm0.0}$ | $40.0_{\pm2.4}$ | $35.0_{\pm2.9}$ | $53.3_{\pm1.7}$ | $50.0_{\pm0.0}$ | $\mathbf{76.7_{\pm1.7}}$ |
| | | 50% | $0.0_{\pm0.0}$ | $38.3_{\pm1.7}$ | $45.0_{\pm0.0}$ | $65.0_{\pm0.0}$ | $53.3_{\pm1.7}$ | $0.0_{\pm0.0}$ | $53.3_{\pm1.7}$ | $46.7_{\pm1.7}$ | $45.0_{\pm0.0}$ | $\mathbf{78.3_{\pm4.4}}$ |
| | | 70% | $0.0_{\pm0.0}$ | $55.0_{\pm0.0}$ | $38.3_{\pm3.3}$ | $65.0_{\pm0.0}$ | $50.0_{\pm0.0}$ | $26.7_{\pm5.4}$ | $30.0_{\pm5.8}$ | $33.3_{\pm1.7}$ | $33.3_{\pm1.7}$ | $\mathbf{66.7_{\pm1.7}}$ |
| | | 90% | $0.0_{\pm0.0}$ | $21.7_{\pm1.7}$ | $26.7_{\pm1.7}$ | $46.7_{\pm1.7}$ | $3.3_{\pm1.7}$ | $0.0_{\pm0.0}$ | $6.7_{\pm3.3}$ | $8.3_{\pm1.7}$ | $15.0_{\pm0.0}$ | $\mathbf{53.3_{\pm1.7}}$ |
| | LP-variant | 0% | $48.3_{\pm3.3}$ | $40.0_{\pm2.9}$ | $41.7_{\pm1.7}$ | $65.0_{\pm0.0}$ | $58.3_{\pm1.7}$ | $30.0_{\pm0.0}$ | $61.7_{\pm1.7}$ | $75.0_{\pm2.9}$ | $71.7_{\pm1.7}$ | $\mathbf{75.0_{\pm0.0}}$ |
| | | 10% | $0.0_{\pm0.0}$ | $36.7_{\pm1.7}$ | $40.0_{\pm0.0}$ | $63.3_{\pm3.3}$ | $60.0_{\pm0.0}$ | $36.7_{\pm2.7}$ | $65.0_{\pm2.9}$ | $70.0_{\pm2.9}$ | $73.3_{\pm1.7}$ | $\mathbf{80.0_{\pm2.9}}$ |
| | | 30% | $0.0_{\pm0.0}$ | $48.3_{\pm3.3}$ | $40.0_{\pm2.9}$ | $60.0_{\pm0.0}$ | $55.0_{\pm0.0}$ | $40.0_{\pm7.1}$ | $53.3_{\pm6.0}$ | $65.0_{\pm2.9}$ | $65.0_{\pm0.0}$ | $\mathbf{75.0_{\pm0.0}}$ |
| | | 50% | $0.0_{\pm0.0}$ | $65.0_{\pm0.0}$ | $35.0_{\pm2.9}$ | $63.3_{\pm3.3}$ | $60.0_{\pm0.0}$ | $38.3_{\pm3.6}$ | $48.3_{\pm4.4}$ | $61.7_{\pm1.7}$ | $65.0_{\pm0.0}$ | $\mathbf{78.3_{\pm1.7}}$ |
| | | 70% | $0.0_{\pm0.0}$ | $46.7_{\pm3.3}$ | $35.0_{\pm0.0}$ | $70.0_{\pm0.0}$ | $60.0_{\pm0.0}$ | $25.0_{\pm8.2}$ | $60.0_{\pm5.0}$ | $56.7_{\pm1.7}$ | $56.7_{\pm1.7}$ | $\mathbf{71.7_{\pm1.7}}$ |
| | | 90% | $0.0_{\pm0.0}$ | $35.0_{\pm2.9}$ | $50.0_{\pm0.0}$ | $65.0_{\pm2.9}$ | $0.0_{\pm0.0}$ | $30.0_{\pm12.5}$ | $50.0_{\pm2.9}$ | $38.3_{\pm1.7}$ | $55.0_{\pm2.9}$ | $\mathbf{66.7_{\pm1.7}}$ |
| Re-mapped label tasks | AG News Remap | 0% | $20.0_{\pm2.9}$ | $15.0_{\pm0.0}$ | $26.7_{\pm1.7}$ | $43.3_{\pm1.7}$ | $43.3_{\pm3.3}$ | $5.0_{\pm2.4}$ | $25.0_{\pm5.0}$ | $40.0_{\pm0.0}$ | $40.0_{\pm0.0}$ | $\mathbf{50.0_{\pm0.0}}$ |
| | | 10% | $5.0_{\pm0.0}$ | $15.0_{\pm0.0}$ | $15.0_{\pm0.0}$ | $41.7_{\pm1.7}$ | $38.3_{\pm1.7}$ | $3.3_{\pm1.4}$ | $26.7_{\pm3.3}$ | $36.7_{\pm1.7}$ | $40.0_{\pm0.0}$ | $\mathbf{51.7_{\pm1.7}}$ |
| | | 30% | $10.0_{\pm0.0}$ | $5.0_{\pm0.0}$ | $5.0_{\pm0.0}$ | $40.0_{\pm0.0}$ | $36.7_{\pm1.7}$ | $1.7_{\pm1.4}$ | $10.0_{\pm0.0}$ | $40.0_{\pm0.0}$ | $43.3_{\pm1.7}$ | $\mathbf{55.0_{\pm0.0}}$ |
| | | 50% | $5.0_{\pm0.0}$ | $10.0_{\pm0.0}$ | $5.0_{\pm0.0}$ | $43.3_{\pm1.7}$ | $35.0_{\pm0.0}$ | $3.3_{\pm1.4}$ | $20.0_{\pm5.0}$ | $35.0_{\pm0.0}$ | $35.0_{\pm0.0}$ | $\mathbf{55.0_{\pm2.9}}$ |
| | | 70% | $5.0_{\pm0.0}$ | $25.0_{\pm0.0}$ | $8.3_{\pm1.7}$ | $50.0_{\pm0.0}$ | $35.0_{\pm0.0}$ | $1.7_{\pm1.4}$ | $11.7_{\pm6.7}$ | $38.3_{\pm1.7}$ | $46.7_{\pm1.7}$ | $\mathbf{58.3_{\pm3.3}}$ |
| | | 90% | $5.0_{\pm0.0}$ | $18.3_{\pm1.7}$ | $5.0_{\pm0.0}$ | $40.0_{\pm0.0}$ | $10.0_{\pm0.0}$ | $15.0_{\pm6.2}$ | $35.0_{\pm0.0}$ | $35.0_{\pm0.0}$ | $41.7_{\pm1.7}$ | $\mathbf{53.3_{\pm1.7}}$ |
| | SST5 Reverse | 0% | $20.0_{\pm0.0}$ | $10.0_{\pm0.0}$ | $13.3_{\pm1.7}$ | $40.0_{\pm0.0}$ | $40.0_{\pm0.0}$ | $15.0_{\pm2.4}$ | $33.3_{\pm6.7}$ | $35.0_{\pm2.9}$ | $40.0_{\pm0.0}$ | $\mathbf{50.0_{\pm2.9}}$ |
| | | 10% | $16.7_{\pm1.7}$ | $10.0_{\pm0.0}$ | $15.0_{\pm0.0}$ | $48.3_{\pm1.7}$ | $40.0_{\pm0.0}$ | $13.3_{\pm2.7}$ | $23.3_{\pm6.7}$ | $33.3_{\pm3.3}$ | $40.0_{\pm0.0}$ | $\mathbf{48.3_{\pm1.7}}$ |
| | | 30% | $23.3_{\pm1.7}$ | $6.7_{\pm1.7}$ | $25.0_{\pm2.9}$ | $40.0_{\pm0.0}$ | $40.0_{\pm0.0}$ | $21.7_{\pm3.6}$ | $26.7_{\pm1.7}$ | $30.0_{\pm0.0}$ | $31.7_{\pm1.7}$ | $\mathbf{46.7_{\pm3.3}}$ |
| | | 50% | $21.7_{\pm1.7}$ | $15.0_{\pm0.0}$ | $15.0_{\pm0.0}$ | $43.3_{\pm1.7}$ | $33.3_{\pm1.7}$ | $21.7_{\pm1.4}$ | $23.3_{\pm1.7}$ | $28.3_{\pm1.7}$ | $30.0_{\pm0.0}$ | $\mathbf{46.7_{\pm3.3}}$ |
| | | 70% | $25.0_{\pm0.0}$ | $23.3_{\pm1.7}$ | $23.3_{\pm1.7}$ | $40.0_{\pm0.0}$ | $30.0_{\pm0.0}$ | $20.0_{\pm2.4}$ | $25.0_{\pm2.9}$ | $36.7_{\pm1.7}$ | $36.7_{\pm1.7}$ | $\mathbf{45.0_{\pm5.0}}$ |
| | | 90% | $20.0_{\pm0.0}$ | $15.0_{\pm2.9}$ | $20.0_{\pm0.0}$ | $30.0_{\pm0.0}$ | $30.0_{\pm0.0}$ | $13.3_{\pm2.7}$ | $21.7_{\pm1.7}$ | $30.0_{\pm0.0}$ | $30.0_{\pm0.0}$ | $\mathbf{31.7_{\pm1.7}}$ |

vantages in settings with noisy data, which will be discussed later in this section.

Another example of our proposed rule-based tasks is constructed by changing the rules for language puzzles (**LP**). A prominent example is "pig Latin" which follows two simple rules: (a) For words that begin with a vowel, one just adds "yay" to the end; (b) for words that begin with consonants, the initial consonants are moved to the end of the word, then "ay" is added. For example, translating "Hello, how are you today?" to pig Latin gives "Ellohay, owhay areyay ouyay odaytay?". Note that this task also requires the model to reason about the language translation rules (i.e., rules (a) and (b)) before predicting for the test sample. The rules can be freely modified (e.g., by changing the suffix word "yay" to others) to create diverse tasks that require in-context reasoning from the LLM. Hence, we create a new variant of the LP task, named **LP-variant**, by using the "ay" suffix for both rules (a) and (b). An example of the task query is given in Example 3 (see App. C.1). As shown in Tab. 2, in this task, our EASE also demonstrates competitive results and outperforms all baselines.

**Remapped label tasks.** By remapping the labels of existing classification tasks to new ones, we construct novel tasks that are against the model's existing knowledge. In the commonly used AG News dataset, the news articles are classified into four categories: "World", "Sports", "Business" and "Sci/Tech". We construct a remapped dataset **AG News Remap** such that "World" news is now labeled as "Sports" news, "Sports" is now labeled as "Business", etc. This is against the LLM's knowledge since the output now does

not correspond to the context descriptions of the input (i.e., news articles). So, the LLM has to learn these remapping rules from the in-context exemplars. Similarly, we construct another dataset **SST5 Reversed** by reversing the labels of the sentiment analysis dataset SST5, such that "very negative" labels are swapped with "very positive" labels, and "negative" labels are swapped with "positive" labels. The results for these novel remapped label tasks are also in Tab. 2, which shows superior performances of EASE over baselines by 6.7%-10% in absolute accuracy.

**Noisy tasks.** Exemplar selection is even more important to achieve good ICL performances when the dataset is potentially noisy, since using noisy or mislabeled data as exemplars could have detrimental effects on performance. Noisy datasets also better resemble practical scenarios because clean data is expensive and difficult to obtain. We construct noisy datasets by injecting various ratios of noisy outputs into the task datasets, ranging from having 10% noisy data samples (i.e., the remaining 90% are clean) to having 90% noisy data samples (i.e., the remaining 10% are clean). We show in Tab. 2 that EASE is the best-performing method across different noise ratios and generally exhibits a lower decrease in accuracy as the tasks become more difficult with increasing noise ratios.

Note that the conclusions on test accuracy results (in App. D.11) are consistent with the main text. These tasks above represent new families of tasks that contain novel knowledge for LLMs. We show through comprehensive experiments that exemplar selection is important and useful in practice, especially for novel downstream task adapta-

Table 3: Average accuracy $\pm$ s.e. for `EASE` with and without jointly optimized instructions. We removed tasks with 100% accuracy. The full results are in App D, Tab. 6.

| | EASE | EASE with instructions | Improve-ment |
|---|---|---|---|
| antonyms | **90.0**$_{\pm0.0}$ | 85.0$_{\pm0.0}$ | -5.0 $\downarrow$ |
| auto_categorization | 30.0$_{\pm0.0}$ | **56.7**$_{\pm1.7}$ | 26.7 $\uparrow$ |
| negation | 95.0$_{\pm0.0}$ | **100.0**$_{\pm0.0}$ | 5.0 $\uparrow$ |
| object_counting | 73.3$_{\pm1.7}$ | **75.0**$_{\pm0.0}$ | 1.7 $\uparrow$ |
| orthography_starts_with | 80.0$_{\pm0.0}$ | **81.7**$_{\pm1.7}$ | 1.7 $\uparrow$ |
| second_word_letter | 53.3$_{\pm1.7}$ | **100.0**$_{\pm0.0}$ | 46.7 $\uparrow$ |
| sentence_similarity | 56.7$_{\pm1.7}$ | **58.3**$_{\pm1.7}$ | 1.7 $\uparrow$ |
| synonyms | 30.0$_{\pm0.0}$ | **31.7**$_{\pm1.7}$ | 1.7 $\uparrow$ |
| taxonomy_animal | 88.3$_{\pm1.7}$ | **100.0**$_{\pm0.0}$ | 11.7 $\uparrow$ |
| translation_en-de | **90.0**$_{\pm0.0}$ | **90.0**$_{\pm0.0}$ | 0.0 $\circ$ |
| translation_en-fr | **88.3**$_{\pm1.7}$ | 85.0$_{\pm0.0}$ | -3.3 $\downarrow$ |
| word_sorting | 90.0$_{\pm0.0}$ | **93.3**$_{\pm1.7}$ | 3.3 $\uparrow$ |
| word_unscrambling | 78.3$_{\pm1.7}$ | **80.0**$_{\pm0.0}$ | 1.7 $\uparrow$ |
| LR (10% noise) | **73.3**$_{\pm1.7}$ | 45.0$_{\pm15.0}$ | -28.3 $\downarrow$ |
| LP-variant (10% noise) | 80.0$_{\pm2.9}$ | **86.7**$_{\pm1.7}$ | 6.7 $\uparrow$ |
| AG News Remap (10% noise) | 51.7$_{\pm1.7}$ | **65.0**$_{\pm0.0}$ | 13.3 $\uparrow$ |
| SST5 Reverse (10% noise) | 48.3$_{\pm1.7}$ | **53.3**$_{\pm1.7}$ | 5.0 $\uparrow$ |

tions. Also, the noisy datasets considered here align with real-world scenarios, which typically contain noises from observation, labeling error, corruption, etc.

### 4.4. Jointly optimizing instruction and exemplars

To the best of our knowledge, no prior work can jointly optimize instruction and exemplars. For a fair comparison, we do not include an instruction when comparing with other exemplar selection baselines in earlier sections. As `EASE` is readily extended to find the optimal combination of instruction and exemplars (Sec. 3.3), we show the benefits of joint optimization in this section. According to Tab. 3, jointly optimizing these two essential components of a prompt significantly improves the performance for most tasks (marked with red arrows $\uparrow$). This improvement is attributed to the ability of the optimized instruction to significantly reinforce the information captured in the corresponding exemplars. For example, an automatically optimized instruction "identify and list the animals from the given words" complements the exemplars in the taxonomy animal task. Therefore, our joint optimization of instruction and exemplars presents a *fully automated pipeline* for prompt optimization and achieves impressive practical performances.

### 4.5. Ablation studies

**Combination with retrieval-based methods to handle larger sets of exemplars.** Applying `EASE` to scenarios where the size $n$ of the set of exemplars $D$ is very large may lead to performance degradation. This is because the space $\Omega$ of exemplar sequences becomes excessively large, which cannot be sufficiently explored without significantly increasing the computation (i.e., with the size of $Q_t$ being fixed). To resolve this issue, a natural idea is to first filter the large pool of data exemplars to eliminate those that are less relevant to the task. To this end, we propose to first

Table 4: Average accuracy $\pm$ s.e. achieved by `EASE` and `EASE` with retrieval for larger exemplar set sizes.

| AG News Remap (10% noise) | | | SST5 Reverse (10% noise) | | |
|---|---|---|---|---|---|
| Size $n$ | EASE | EASE with retrieval | Size $n$ | EASE | EASE with retrieval |
| 1000 | 50.0$_{\pm2.9}$ | **60.0**$_{\pm0.0}$ | 1000 | 43.3$_{\pm3.3}$ | **48.3**$_{\pm1.7}$ |
| 10000 | 55.0$_{\pm0.0}$ | **63.3**$_{\pm1.7}$ | 3000 | 48.3$_{\pm4.4}$ | **48.3**$_{\pm1.7}$ |
| 50000 | 48.3$_{\pm1.7}$ | **63.3**$_{\pm1.7}$ | 5000 | 43.3$_{\pm1.7}$ | **50.0**$_{\pm0.0}$ |
| 100000 | 50.0$_{\pm2.9}$ | **65.0**$_{\pm0.0}$ | 7000 | 45.0$_{\pm0.0}$ | **48.3**$_{\pm1.7}$ |

use retrieval-based methods to select the exemplars that are more relevant to the task, and then run our `EASE` using this refined smaller set of exemplars. Specifically, we use a cosine similarity retriever and perform exemplar selection on $D$ with a size $n$ as large as 100000. As shown in Tab. 4, when the size $n$ of the exemplar set is large, combining `EASE` with retrieval gives better performances than directly running `EASE`.

**Effectiveness of components of `EASE`.** Here we show the necessity of both of the main components to the success of `EASE`: OT and NeuralUCB. As shown in Tab. 7 of App. D.3, `EASE` significantly outperforms methods employing OT or NeuralUCB alone. Overall, a pure subset selection algorithm based on OT performs badly on its own, especially when the dataset's noise ratio is high. However, when used together with NeuralUCB, OT significantly improves the performance of our `EASE`.

**Further ablations.** We defer further ablation studies to App. D, including those exploring (a) speedups from OT, (b) a larger $k$, (c) benefits of using an ordering-aware embedding, (d) different black-box LLMs, (e) other embedding models, (f) degree of exploration $\nu$, and (g) asking GPT to directly select exemplars.

## 5. Conclusion and limitation

We propose `EASE`, an algorithm that selects the optimal ordered set of exemplars for in-context learning of black-box LLMs in an automated fashion. `EASE` is query-efficient due to the adoption of the NeuralUCB algorithm and is further made computationally feasible for large spaces of exemplar sequences through a technique based on optimal transport. Additionally, our `EASE` has been extended to a fully automated prompt optimization pipeline that jointly optimizes exemplars and instruction for the best in-context learning performance. Furthermore, we provide practical insights indicating that exemplar selection in in-context learning is more crucial for downstream tasks that the LLM has limited knowledge about. However, the on-the-fly computation of embedding for the ordered exemplar sequences is the computational bottleneck of our method, which could be potentially improved in future work for more efficient optimization. Also, a potential limitation of `EASE` is the requirement for a suitable validation set, which may not be readily available in some scenarios.

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

# A. Related work

**Retrieval-based methods.** This approach utilizes trainable exemplar retrievers to select exemplars depending on the text sequence of each individual test sample (Ye et al., 2023; Albalak et al., 2024). Liu et al. (2022) first proposed a similarity-based (e.g., negative Euclidean distance or cosine similarity) retrieval strategy on the sentence embeddings of the exemplars for ICL. Gao et al. (2024) further enhanced the above by emphasizing on exemplars with top-2 labels that the LLM is most uncertain about. Adapting retrievers to specific tasks, Rubin et al. (2022) learned a retrieval model from evaluating individual exemplars by their 1-shot ICL performance on validation data. Improving on the limitation of the above methods that exemplars are considered in isolation (i.e., independently), Ye et al. (2023) proposed to retrieve a set of exemplars jointly by examining their joint probability to capture inter-relationships. Likewise, Levy et al. (2023) retrieved a set by selecting diverse exemplars that collectively cover all of the output structures. Gupta et al. (2023) instead generalized the individual metrics to a set-level metric through a submodular function, which is well-suited for optimization via greedy algorithms. However, retrieval-based methods are characterized by varying exemplars for each test sample, which do not align with our practical setting of maintaining a fixed set of exemplars for the entire task. Also, another common drawback is that heuristics are typically required to order the exemplars in the retrieved set.

**Selection of a fixed set of exemplars.** Having a fixed set of exemplars offers practical and privacy-related advantages, such as the ease of implementation and reduced data exposure (Luo et al., 2024; Neel & Chang, 2024). Wang et al. (2023) proposed to train a small LLM as a latent variable model using output token logits from independent exemplars, then forming a fixed exemplar set to directly transfer to target models. Similarly, Chang & Jia (2023) developed a data model for each validation sample and introduced an aggregate metric to select subsets. Li & Qiu (2023) proposed to filter and search for best exemplars using an informativeness metric derived from LLM output logits on classification tasks. However, these works are restricted to classification tasks. The closest to our work is that of Zhang et al. (2022), which used reinforcement learning to actively select exemplars. In a similar setting to ours, Nguyen & Wong (2023) used influence to select the most influential exemplars and order them arbitrarily to form the subset. However, both methods perform worse than our algorithm in our experiments.

**Instruction optimization.** Another related direction for enhancing the performance of LLMs is instruction optimization. Focusing on instructions within the prompt, evolutionary algorithms and zeroth-order optimization algorithms are gaining popularity in refining the prompts for black-box LLMs (Chen et al., 2023a; Lin et al., 2023; Guo et al., 2024; Fernando et al., 2023; Yang et al., 2024; Hu et al., 2024). Specifically, some studies (Lin et al., 2023; Fernando et al., 2023; Yang et al., 2024; Hu et al., 2024) maintained a constant set of exemplars throughout the process of prompt optimization, whereas others (Chen et al., 2023a; Guo et al., 2024) ignored the consideration of exemplars altogether by adopting a zero-shot setting. It is therefore imperative to develop an effective exemplar selection method for black-box LLMs.

# B. Broader impacts

As LLMs continue to gain popularity and are increasingly used by a broad user base (e.g., evidenced by OpenAI's over 1 million active users), carefully handling the ethical considerations associated with LLMs' diverse applications becomes crucial. Such a work like ours which focuses on automatically optimizing the performance of LLMs offers valuable applications and ease of usage. However, it is both challenging and essential to address the potential for malicious usage. When applied to tasks specially designed with malicious or adversarial intent, our method could be exploited to produce harmful instructions and exemplars. In such cases, responsible usage of the tool is important. There may be a need for a user-friendly platform (which integrates safety measures, instead of providing the raw code) to add an extra layer of protection. Additionally, we urge the community to focus more on potential ethical and safety issues related to the usage of LLMs and associated technologies, which should also account for additional objectives and constraints (e.g., harmfulness).

# C. Implementation details

In this section, we provide details for the implementation of the data processing (see App. C.1) and the baseline methods employed in this paper (see App. C.2). All experiments are conducted on a server with Intel(R) Xeon(R) CPU and NVIDIA H100 GPUs. Unless otherwise stated, we use "gpt-3.5-turbo-1106" as the target black-box model, and MPNet as the embedding model.

## C.1. Details for data processing

We follow the query template proposed by APE (Zhou et al., 2023) for querying LLM models. Example 1 below shows the general query template for in-context learning (ICL) used in the paper, and the LLM will generate outputs corresponding to the query. In the template, the placeholders (i.e., [INSTRUCTION], [INPUT], [OUTPUT] and [TEST INPUT]) are replaced by raw text in the datasets. The placeholder "<More exemplars...>" represents any additional exemplars written the same input-output format depending on the number of in-context exemplars $k$ in the prompt. The "instruction" is optional and we only include the instruction when jointly optimizing for both the exemplars and the instruction (Sec. 3.3 and Sec. 4.4).

---

**Example Query 1: A General Template**

*(optional)* Instruction: [INSTRUCTION]

Input: [INPUT]
Output: [OUTPUT]

<More exemplars...>

Input: [INPUT]
Output: [OUTPUT]

Input: [TEST INPUT]
Output:

---

Next, we present the details for the new families of out-of-distribution tasks that we proposed in Sec. 4.3.

**Linear regression (LR)**. The LR task is generated with an underlying linear regression function $y = ax + b$, where $a, b$ are the coefficients to be defined. In our specific example of the LR task presented in the paper, we arbitrarily choose $a = -4$ and $b = 6$. We let $x$ be the input and $y$ be the output. Note that to make the task difficult, no information about the function structure (i.e., $y = ax + b$) is passed to the LLM in the query. A concrete example is shown in Example 2.

**Language puzzle variant (LP-variant)**. We change the rules for the classic language game "pig Latin". The original pig Latin follows 2 rules: (a) For words that begin with a vowel, one just adds "yay" to the end; (b) For words that begin with consonants, the initial consonants are moved to the end of the word, then "ay" is added. We create a variant, LP-variant, using the "ay" suffix for both rules (a) and (b). An example of the task query is given in Example 3. Note that one could freely modify the rules, by changing the suffix word "yay" to something else for example, and create diverse tasks that require in-context reasoning from the model.

---

**Example Query 2: LR**

Input: 172
Output: -682

<More exemplars...>

Input: 47
Output: -182

Input: 117
Output:

---

**Example Query 3: LP-variant**

Input: quick brown fox The jumps Over the Lazy dog
Output: uickqay ownbray oxfay Ethay umpsjay Overyay ethay Azylay ogday

<More exemplars...>

Input: Tom never walks to school
Output: Omtay evernay alksway otay oolschay

Input: We never talk
Output:

---

**AG News Remap**. For the commonly-used AG News dataset, the news articles are classified into four categories: "World", "Sports", "Business" and "Sci/Tech". We constructed a re-mapped dataset by "shifting" the label mapping, such that

- "World" news is now labeled as "Sports" news,

- "Sports" news is now labeled as "Business" news,

- "Business" news is now labeled as "Sci/Tech" news,

- "Sci/Tech" news is now labeled as "World" news.

Note that the remapping rule above is chosen arbitrarily. Therefore, one could create a variety of tasks by changing the remapping rule, or even directly changing the labels completely. For the above rule that we adopted, an example of the AG News Remap task query is given in Example 4.

---

**Example Query 4: AG News Remap**

Input: Yahoo, Adobe join hands Adobe Systems, the digital imaging, design and document technology platform provider and Internet service provider Yahoo will announce this week the launch of a co-branded Yahoo Toolbar.
Output: World

<More exemplars...>

Input: Schumacher Sets Mark Michael Schumacher won the Hungarian Grand Prix Sunday in Budapest, setting yet another record by becoming the first Formula One driver with 12 victories in a season.
Output: Business

Input: LeapFrog Warns on 3Q, Year Profit View LeapFrog Enterprises Inc., a developer of technology-based educational products, on Monday lowered third-quarter and full-year profit expectations, citing difficult market conditions.
Output:

---

**SST5 Reverse**. SST5 is another commonly used 5-way sentiment classification dataset, with labels being "very positive", "positive", "neutral", "negative" and "very negative". To make the task novel and unseen by the LLM before, we reverse the labels to be against the sentiment expressed in the input, such that

- "very positive" sentences are now labeled as "very negative",

- "positive" sentences are now labeled as "negative",

- "neutral" sentences are still labeled as "neutral",

- "negative" sentences are now labeled as "positive",

- "very negative" sentences are now labeled as "very positive".

This makes the task difficult as the LLM has to now output sentiments that are against the pre-trained knowledge about sentiments, which is obtained from the large corpus of pre-training data that the LLM has been trained on.

An example of the SST5 Reverse task query is given in Example 5.

---

**Example Query 5: SST5 Reverse**

Input: extremely bad .
Output: very positive

<More exemplars...>

Input: ... bright , intelligent , and humanly funny film .
Output: very negative

Input: really dumb but occasionally really funny .
Output:

---

Table 5: Average validation accuracy $\pm$ standard error achieved by the best exemplar sequence discovered by different algorithms for different tasks over 3 independent trials.

| | DPP | MMD | OT | Cosine | BM25 | Active | Inf | Evo | Best-of-N | EASE |
|---|---|---|---|---|---|---|---|---|---|---|
| active_to_passive | $100.0_{\pm0.0}$ | $100.0_{\pm0.0}$ | $100.0_{\pm0.0}$ | $100.0_{\pm0.0}$ | $100.0_{\pm0.0}$ | $100.0_{\pm0.0}$ | $100.0_{\pm0.0}$ | $100.0_{\pm0.0}$ | $100.0_{\pm0.0}$ | $100.0_{\pm0.0}$ |
| antonyms | $70.0_{\pm0.0}$ | $80.0_{\pm0.0}$ | $81.7_{\pm1.7}$ | $85.0_{\pm0.0}$ | $85.0_{\pm0.0}$ | $80.0_{\pm0.0}$ | $86.7_{\pm1.7}$ | $88.3_{\pm1.7}$ | $90.0_{\pm0.0}$ | $90.0_{\pm0.0}$ |
| auto_categorization | $3.3_{\pm1.7}$ | $8.3_{\pm1.7}$ | $0.0_{\pm0.0}$ | $25.0_{\pm0.0}$ | $16.7_{\pm1.7}$ | $10.0_{\pm2.4}$ | $21.7_{\pm1.7}$ | $21.7_{\pm1.7}$ | $20.0_{\pm0.0}$ | $30.0_{\pm0.0}$ |
| diff | $0.0_{\pm0.0}$ | $0.0_{\pm0.0}$ | $0.0_{\pm0.0}$ | $0.0_{\pm0.0}$ | $0.0_{\pm0.0}$ | $0.0_{\pm0.0}$ | $100.0_{\pm0.0}$ | $100.0_{\pm0.0}$ | $100.0_{\pm0.0}$ | $100.0_{\pm0.0}$ |
| first_word_letter | $100.0_{\pm0.0}$ | $100.0_{\pm0.0}$ | $100.0_{\pm0.0}$ | $100.0_{\pm0.0}$ | $100.0_{\pm0.0}$ | $100.0_{\pm0.0}$ | $100.0_{\pm0.0}$ | $100.0_{\pm0.0}$ | $100.0_{\pm0.0}$ | $100.0_{\pm0.0}$ |
| larger_animal | $70.0_{\pm0.0}$ | $91.7_{\pm1.7}$ | $100.0_{\pm0.0}$ | $100.0_{\pm0.0}$ | $100.0_{\pm0.0}$ | $66.7_{\pm1.4}$ | $100.0_{\pm0.0}$ | $100.0_{\pm0.0}$ | $100.0_{\pm0.0}$ | $100.0_{\pm0.0}$ |
| letters_list | $100.0_{\pm0.0}$ | $100.0_{\pm0.0}$ | $100.0_{\pm0.0}$ | $100.0_{\pm0.0}$ | $100.0_{\pm0.0}$ | $100.0_{\pm0.0}$ | $100.0_{\pm0.0}$ | $100.0_{\pm0.0}$ | $100.0_{\pm0.0}$ | $100.0_{\pm0.0}$ |
| negation | $95.0_{\pm0.0}$ | $95.0_{\pm0.0}$ | $95.0_{\pm0.0}$ | $95.0_{\pm0.0}$ | $95.0_{\pm0.0}$ | $95.0_{\pm0.0}$ | $95.0_{\pm0.0}$ | $95.0_{\pm0.0}$ | $95.0_{\pm0.0}$ | $95.0_{\pm0.0}$ |
| num_to_verbal | $100.0_{\pm0.0}$ | $100.0_{\pm0.0}$ | $100.0_{\pm0.0}$ | $100.0_{\pm0.0}$ | $100.0_{\pm0.0}$ | $100.0_{\pm0.0}$ | $100.0_{\pm0.0}$ | $100.0_{\pm0.0}$ | $100.0_{\pm0.0}$ | $100.0_{\pm0.0}$ |
| object_counting | $55.0_{\pm2.9}$ | $56.7_{\pm1.7}$ | $48.3_{\pm1.7}$ | $61.7_{\pm1.7}$ | $66.7_{\pm1.7}$ | $51.7_{\pm1.4}$ | $63.3_{\pm4.4}$ | $70.0_{\pm0.0}$ | $70.0_{\pm0.0}$ | $73.3_{\pm1.7}$ |
| orthography_starts_with | $20.0_{\pm2.9}$ | $35.0_{\pm0.0}$ | $61.7_{\pm1.7}$ | $78.3_{\pm1.7}$ | $70.0_{\pm0.0}$ | $43.3_{\pm1.4}$ | $70.0_{\pm2.9}$ | $75.0_{\pm0.0}$ | $78.3_{\pm1.7}$ | $80.0_{\pm0.0}$ |
| rhymes | $60.0_{\pm0.0}$ | $51.7_{\pm1.7}$ | $0.0_{\pm0.0}$ | $100.0_{\pm0.0}$ | $80.0_{\pm0.0}$ | $65.0_{\pm8.2}$ | $70.0_{\pm13.2}$ | $100.0_{\pm0.0}$ | $100.0_{\pm0.0}$ | $100.0_{\pm0.0}$ |
| second_word_letter | $10.0_{\pm2.9}$ | $30.0_{\pm0.0}$ | $28.3_{\pm1.7}$ | $50.0_{\pm0.0}$ | $50.0_{\pm0.0}$ | $26.7_{\pm8.3}$ | $40.0_{\pm0.0}$ | $46.7_{\pm1.7}$ | $50.0_{\pm0.0}$ | $53.3_{\pm1.7}$ |
| sentence_similarity | $20.0_{\pm0.0}$ | $21.7_{\pm3.3}$ | $40.0_{\pm2.9}$ | $46.7_{\pm1.7}$ | $53.3_{\pm1.7}$ | $5.0_{\pm4.1}$ | $18.3_{\pm6.7}$ | $45.0_{\pm0.0}$ | $51.7_{\pm1.7}$ | $56.7_{\pm1.7}$ |
| sentiment | $85.0_{\pm0.0}$ | $90.0_{\pm0.0}$ | $85.0_{\pm0.0}$ | $96.7_{\pm1.7}$ | $100.0_{\pm0.0}$ | $85.0_{\pm4.1}$ | $91.7_{\pm1.7}$ | $100.0_{\pm0.0}$ | $100.0_{\pm0.0}$ | $100.0_{\pm0.0}$ |
| singular_to_plural | $100.0_{\pm0.0}$ | $100.0_{\pm0.0}$ | $100.0_{\pm0.0}$ | $100.0_{\pm0.0}$ | $100.0_{\pm0.0}$ | $100.0_{\pm0.0}$ | $100.0_{\pm0.0}$ | $100.0_{\pm0.0}$ | $100.0_{\pm0.0}$ | $100.0_{\pm0.0}$ |
| sum | $0.0_{\pm0.0}$ | $0.0_{\pm0.0}$ | $0.0_{\pm0.0}$ | $0.0_{\pm0.0}$ | $0.0_{\pm0.0}$ | $0.0_{\pm0.0}$ | $100.0_{\pm0.0}$ | $100.0_{\pm0.0}$ | $100.0_{\pm0.0}$ | $100.0_{\pm0.0}$ |
| synonyms | $10.0_{\pm0.0}$ | $25.0_{\pm0.0}$ | $20.0_{\pm0.0}$ | $35.0_{\pm0.0}$ | $30.0_{\pm0.0}$ | $3.3_{\pm1.4}$ | $26.7_{\pm1.7}$ | $30.0_{\pm0.0}$ | $30.0_{\pm0.0}$ | $30.0_{\pm0.0}$ |
| taxonomy_animal | $43.3_{\pm4.4}$ | $40.0_{\pm2.9}$ | $46.7_{\pm1.7}$ | $85.0_{\pm2.9}$ | $80.0_{\pm0.0}$ | $45.0_{\pm6.2}$ | $70.0_{\pm5.0}$ | $80.0_{\pm0.0}$ | $80.0_{\pm0.0}$ | $88.3_{\pm1.7}$ |
| translation_en-de | $90.0_{\pm0.0}$ | $80.0_{\pm0.0}$ | $80.0_{\pm0.0}$ | $90.0_{\pm0.0}$ | $85.0_{\pm0.0}$ | $56.7_{\pm13.0}$ | $90.0_{\pm0.0}$ | $90.0_{\pm0.0}$ | $90.0_{\pm0.0}$ | $90.0_{\pm0.0}$ |
| translation_en-es | $90.0_{\pm0.0}$ | $100.0_{\pm0.0}$ | $96.7_{\pm1.7}$ | $100.0_{\pm0.0}$ | $100.0_{\pm0.0}$ | $96.7_{\pm1.4}$ | $98.3_{\pm1.7}$ | $100.0_{\pm0.0}$ | $100.0_{\pm0.0}$ | $100.0_{\pm0.0}$ |
| translation_en-fr | $76.7_{\pm1.7}$ | $76.7_{\pm1.7}$ | $81.7_{\pm1.7}$ | $85.0_{\pm0.0}$ | $85.0_{\pm0.0}$ | $81.7_{\pm1.4}$ | $85.0_{\pm0.0}$ | $86.7_{\pm1.7}$ | $85.0_{\pm0.0}$ | $88.3_{\pm1.7}$ |
| word_sorting | $26.7_{\pm1.7}$ | $88.3_{\pm1.7}$ | $88.3_{\pm1.7}$ | $90.0_{\pm0.0}$ | $71.7_{\pm1.7}$ | $80.0_{\pm0.0}$ | $88.3_{\pm1.7}$ | $93.3_{\pm1.7}$ | $91.7_{\pm1.7}$ | $90.0_{\pm0.0}$ |
| word_unscrambling | $68.3_{\pm1.7}$ | $56.7_{\pm1.7}$ | $71.7_{\pm1.7}$ | $75.0_{\pm0.0}$ | $76.7_{\pm1.7}$ | $63.3_{\pm3.6}$ | $66.7_{\pm1.7}$ | $75.0_{\pm0.0}$ | $75.0_{\pm0.0}$ | $78.3_{\pm1.7}$ |
| # best-performing tasks | 7 | 7 | 7 | 11 | 9 | 6 | 10 | 14 | 14 | **22** |

**Injecting noises into datasets.** To construct noisy datasets with different noise ratios, we mainly modify the labels of data points. Specifically, to construct a dataset with $r\%$ noisy data, we sample $r\%$ data and replace their labels with the labels of other randomly sampled data points in the dataset. We also design specific noise structures for LR and LP-variant to simulate noises due to a systematic error. For LR, the noisy data instead follows $y = 5x - 8$. For LP-variant, the noisy data simply repeats the input sentence in the label.

## C.2. Details for implementation of the methods and baselines

For all methods, we use a consistent exemplar selection set $D$, validation set $D_V$ and test set for evaluating one task. We also implement all exemplar selection without replacement for all methods.

**DDP**. We follow Ye et al. (2023) to use the DPP metric combined with relevance scores as the subset selection criterion. We do not train the embedding model and directly use cosine similarity to measure the relevance term. However, the original metric by Ye et al. (2023) is dependent on the test input $x_{\text{test}}$, so we adapted the method to average the metric over the validation dataset. The exemplar set with the maximum values on the metric is chosen and evaluated on the black-box LLM.

**MMD**. We use the MMD metric in Sejdinovic et al. (2013) to measure the distance between the exemplars set and the validation set. The exemplar set with the maximum values on the MMD metric is chosen and evaluated on the black-box LLM.

**OT**. We use the OT metric in (4) to measure the distance between the exemplars set and the validation set. Similarly, the exemplar set with the maximum values on the OT metric is chosen and evaluated on the black-box LLM.

**Cosine**. Retrieval-based methods are not suitable for finding a fixed set of exemplars for the entire test set in its original form. Hence, we propose the following adaptation. We first retrieve top-$R$ candidates with the lowest distance to all validation samples on average. Then, we sample $T$ permutations of exemplars from the $R$ candidates to test on the black-box LLM, where $T$ is the black-box query budget. In the above, the retriever calculates the cosine similarity between the embeddings of samples obtained from an embedding model $h(\cdot)$. In this paper, we use $R = 10$ and use MPNet as $h(\cdot)$.

**BM25**. This retrieval-based baseline works similarly to Cosine. The only difference is that this baseline uses the classic BM25 (Robertson & Zaragoza, 2009) as the retriever.

Table 6: Average accuracy $\pm$ standard error comparison for EASE with and without jointly optimized instructions.

| | EASE | EASE with instructions | improvement |
|---|---|---|---|
| antonyms | **90.0**$_{\pm 0.0}$ | 85.0$_{\pm 0.0}$ | -5.0 $\downarrow$ |
| auto_categorization | 30.0$_{\pm 0.0}$ | **56.7**$_{\pm 1.7}$ | 26.7 $\uparrow$ |
| diff | **100.0**$_{\pm 0.0}$ | **100.0**$_{\pm 0.0}$ | 0.0 $\circ$ |
| larger_animal | **100.0**$_{\pm 0.0}$ | **100.0**$_{\pm 0.0}$ | 0.0 $\circ$ |
| negation | 95.0$_{\pm 0.0}$ | **100.0**$_{\pm 0.0}$ | 5.0 $\uparrow$ |
| object_counting | 73.3$_{\pm 1.7}$ | **75.0**$_{\pm 0.0}$ | 1.7 $\uparrow$ |
| orthography_starts_with | 80.0$_{\pm 0.0}$ | **81.7**$_{\pm 1.7}$ | 1.7 $\uparrow$ |
| rhymes | **100.0**$_{\pm 0.0}$ | **100.0**$_{\pm 0.0}$ | 0.0 $\circ$ |
| second_word_letter | 53.3$_{\pm 1.7}$ | **100.0**$_{\pm 0.0}$ | 46.7 $\uparrow$ |
| sentence_similarity | 56.7$_{\pm 1.7}$ | **58.3**$_{\pm 1.7}$ | 1.7 $\uparrow$ |
| sentiment | **100.0**$_{\pm 0.0}$ | **100.0**$_{\pm 0.0}$ | 0.0 $\circ$ |
| sum | **100.0**$_{\pm 0.0}$ | **100.0**$_{\pm 0.0}$ | 0.0 $\circ$ |
| synonyms | 30.0$_{\pm 0.0}$ | **31.7**$_{\pm 1.7}$ | 1.7 $\uparrow$ |
| taxonomy_animal | 88.3$_{\pm 1.7}$ | **100.0**$_{\pm 0.0}$ | 11.7 $\uparrow$ |
| translation_en-de | **90.0**$_{\pm 0.0}$ | **90.0**$_{\pm 0.0}$ | 0.0 $\circ$ |
| translation_en-es | **100.0**$_{\pm 0.0}$ | **100.0**$_{\pm 0.0}$ | 0.0 $\circ$ |
| translation_en-fr | **88.3**$_{\pm 1.7}$ | 85.0$_{\pm 0.0}$ | -3.3 $\downarrow$ |
| word_sorting | 90.0$_{\pm 0.0}$ | **93.3**$_{\pm 1.7}$ | 3.3 $\uparrow$ |
| word_unscrambling | 78.3$_{\pm 1.7}$ | **80.0**$_{\pm 0.0}$ | 1.7 $\uparrow$ |
| LR (10% noise) | **73.3**$_{\pm 1.7}$ | 45.0$_{\pm 15.0}$ | -28.3 $\downarrow$ |
| LP-variant (10% noise) | 80.0$_{\pm 2.9}$ | **86.7**$_{\pm 1.7}$ | 6.7 $\uparrow$ |
| AG News Remap (10% noise) | 51.7$_{\pm 1.7}$ | **65.0**$_{\pm 0.0}$ | 13.3 $\uparrow$ |
| SST5 Reverse (10% noise) | 48.3$_{\pm 1.7}$ | **53.3**$_{\pm 1.7}$ | 5.0 $\uparrow$ |

**Active**. We use the official implementation of Zhang et al. (2022). For a fair comparison using the same query budget $T$ for the black-box LLM, we limit the number of episodes that can be used according to $T$, and then train the exemplar selection policy through reinforcement learning.

**Inf**. For Inf, we first sample random permutations of exemplar sequences to be evaluated on the black-box LLM to obtain the scores. Then, we follow the influence metric proposed by Nguyen & Wong (2023) to select $k$ individual exemplars to form the best exemplar sequence. The best exemplar sequence is then finally evaluated on the black-box LLM.

**Evo.** We propose a new baseline using evolutionary strategies. For each iteration, the mutation operator changes one exemplar in the current best exemplar sequence to another random exemplar. Then, we evaluate the exemplars on the black-box LLM. In this way, we exploit the current knowledge about the best exemplars and perform local mutations.

**Best-of-N.** This is a straightforward baseline that explores the whole space of all exemplar permutations uniformly by sampling random permutations until the $T$ query budget is exhausted.

**EASE**. The details for our proposed method EASE are described thoroughly in Sec. 3. We use a sampling size of $q = 50000$ exemplar permutations per iteration after OT is introduced.

# D. More experiment results

## D.1. Full results table for the 24 tasks

Tab. 5 is the full table containing all 24 tasks (with more than 100 training data samples) from the Instruction Induction dataset (compared to Tab. 1 which drops tasks with 100% accuracy across baselines). Our EASE outperforms all baselines.

## D.2. Full results table for EASE with instructions

We present the full table of results for EASE with joint optimization of exemplars and instructions in Tab. 6 (compared to Tab. 3 which drops tasks with 100% accuracy both with and without instructions). The conclusion is still consistent with the main text that incorporating instructions jointly optimized to best match the chosen exemplars could improve the ICL performance, especially for difficult tasks (e.g., auto categorization, taxonomy animal, etc.).

### D.3. Ablation studies for the NeuralUCB and OT components

The following Tab. 7 and Tab. 8 show the necessity of both of the main components, NeuralUCB and OT, in the success of the proposed EASE algorithm. A pure subset selection algorithm based on OT performs badly on its own, especially when the noise ratio is high in the datasets. However, it greatly improves the performance of NeuralUCB when used together with NeuralUCB in our EASE. This is attributed to the ability to examine more permutations of the exemplars without increasing the computational cost, since only selected permutations through OT (which places an implicit bias towards more relevant exemplars) will be evaluated for the acquisition values in NeuralUCB.

Table 7: Average accuracy $\pm$ standard error for ablation studies of the OT and NeuralUCB components in EASE.

| Task | Noise | OT | NeuralUCB | EASE |
|---|---|---|---|---|
| LR | 0% | $50.0_{\pm0.0}$ | $71.7_{\pm1.7}$ | $\mathbf{80.0_{\pm2.9}}$ |
| | 10% | $48.3_{\pm1.7}$ | $66.7_{\pm3.3}$ | $\mathbf{73.3_{\pm1.7}}$ |
| | 30% | $46.7_{\pm1.7}$ | $66.7_{\pm1.7}$ | $\mathbf{76.7_{\pm1.7}}$ |
| | 50% | $45.0_{\pm0.0}$ | $61.7_{\pm1.7}$ | $\mathbf{78.3_{\pm4.4}}$ |
| | 70% | $38.3_{\pm3.3}$ | $55.0_{\pm5.0}$ | $\mathbf{66.7_{\pm1.7}}$ |
| | 90% | $26.7_{\pm1.7}$ | $28.3_{\pm3.3}$ | $\mathbf{53.3_{\pm1.7}}$ |
| LP-variant | 0% | $41.7_{\pm1.7}$ | $\mathbf{78.3_{\pm1.7}}$ | $75.0_{\pm0.0}$ |
| | 10% | $40.0_{\pm0.0}$ | $75.0_{\pm0.0}$ | $\mathbf{80.0_{\pm2.9}}$ |
| | 30% | $40.0_{\pm2.9}$ | $70.0_{\pm0.0}$ | $\mathbf{75.0_{\pm0.0}}$ |
| | 50% | $35.0_{\pm2.9}$ | $71.7_{\pm1.7}$ | $\mathbf{78.3_{\pm1.7}}$ |
| | 70% | $35.0_{\pm0.0}$ | $70.0_{\pm2.9}$ | $\mathbf{71.7_{\pm1.7}}$ |
| | 90% | $50.0_{\pm0.0}$ | $61.7_{\pm1.7}$ | $\mathbf{66.7_{\pm1.7}}$ |
| AG News Remap | 0% | $26.7_{\pm1.7}$ | $48.3_{\pm1.7}$ | $\mathbf{50.0_{\pm0.0}}$ |
| | 10% | $15.0_{\pm0.0}$ | $51.7_{\pm3.3}$ | $\mathbf{51.7_{\pm1.7}}$ |
| | 30% | $5.0_{\pm0.0}$ | $55.0_{\pm2.9}$ | $\mathbf{55.0_{\pm0.0}}$ |
| | 50% | $5.0_{\pm0.0}$ | $41.7_{\pm3.3}$ | $\mathbf{55.0_{\pm2.9}}$ |
| | 70% | $8.3_{\pm1.7}$ | $45.0_{\pm2.9}$ | $\mathbf{58.3_{\pm3.3}}$ |
| | 90% | $5.0_{\pm0.0}$ | $45.0_{\pm5.8}$ | $\mathbf{53.3_{\pm1.7}}$ |
| SST5 Reverse | 0% | $13.3_{\pm1.7}$ | $41.7_{\pm3.3}$ | $\mathbf{50.0_{\pm2.9}}$ |
| | 10% | $15.0_{\pm0.0}$ | $48.3_{\pm4.4}$ | $\mathbf{48.3_{\pm1.7}}$ |
| | 30% | $25.0_{\pm2.9}$ | $36.7_{\pm4.4}$ | $\mathbf{46.7_{\pm3.3}}$ |
| | 50% | $15.0_{\pm0.0}$ | $38.3_{\pm1.7}$ | $\mathbf{46.7_{\pm3.3}}$ |
| | 70% | $23.3_{\pm1.7}$ | $\mathbf{53.3_{\pm3.3}}$ | $45.0_{\pm5.0}$ |
| | 90% | $20.0_{\pm0.0}$ | $\mathbf{33.3_{\pm3.3}}$ | $31.7_{\pm1.7}$ |

Table 8: Average test accuracy $\pm$ standard error for ablation studies of the OT and NeuralUCB components in EASE.

| Task | Noise | OT | NeuralUCB | EASE |
|---|---|---|---|---|
| LR | 0% | $34.0_{\pm1.0}$ | $50.7_{\pm3.2}$ | $\mathbf{58.0_{\pm2.1}}$ |
| | 10% | $29.3_{\pm0.3}$ | $41.0_{\pm2.9}$ | $\mathbf{42.3_{\pm0.9}}$ |
| | 30% | $30.0_{\pm1.0}$ | $\mathbf{47.7_{\pm2.9}}$ | $47.7_{\pm5.8}$ |
| | 50% | $23.0_{\pm0.6}$ | $49.3_{\pm1.7}$ | $\mathbf{50.7_{\pm2.2}}$ |
| | 70% | $28.7_{\pm0.7}$ | $44.0_{\pm2.1}$ | $\mathbf{47.0_{\pm3.2}}$ |
| | 90% | $32.0_{\pm0.6}$ | $14.7_{\pm2.2}$ | $\mathbf{39.0_{\pm7.8}}$ |
| LP-variant | 0% | $34.7_{\pm0.7}$ | $\mathbf{56.3_{\pm4.1}}$ | $53.0_{\pm2.3}$ |
| | 10% | $34.0_{\pm0.6}$ | $\mathbf{55.7_{\pm4.8}}$ | $48.0_{\pm2.5}$ |
| | 30% | $36.0_{\pm1.1}$ | $48.7_{\pm2.6}$ | $\mathbf{50.7_{\pm1.4}}$ |
| | 50% | $31.0_{\pm0.6}$ | $49.3_{\pm2.6}$ | $\mathbf{56.7_{\pm0.9}}$ |
| | 70% | $30.3_{\pm0.9}$ | $50.3_{\pm3.0}$ | $\mathbf{52.0_{\pm1.0}}$ |
| | 90% | $39.7_{\pm0.3}$ | $38.3_{\pm4.8}$ | $\mathbf{44.3_{\pm1.4}}$ |
| AG News Remap | 0% | $12.3_{\pm0.7}$ | $29.7_{\pm0.9}$ | $\mathbf{30.3_{\pm1.2}}$ |
| | 10% | $13.0_{\pm1.1}$ | $\mathbf{32.0_{\pm4.5}}$ | $30.3_{\pm4.4}$ |
| | 30% | $4.7_{\pm0.3}$ | $32.0_{\pm3.1}$ | $\mathbf{40.0_{\pm1.0}}$ |
| | 50% | $3.7_{\pm0.3}$ | $30.7_{\pm3.7}$ | $\mathbf{40.0_{\pm3.1}}$ |
| | 70% | $7.0_{\pm0.0}$ | $34.3_{\pm1.9}$ | $\mathbf{45.0_{\pm2.0}}$ |
| | 90% | $2.0_{\pm0.0}$ | $26.3_{\pm8.4}$ | $\mathbf{36.7_{\pm3.2}}$ |
| SST5 Reverse | 0% | $9.7_{\pm0.9}$ | $31.3_{\pm5.2}$ | $\mathbf{34.0_{\pm3.0}}$ |
| | 10% | $9.3_{\pm0.3}$ | $\mathbf{37.0_{\pm1.5}}$ | $31.3_{\pm2.6}$ |
| | 30% | $11.7_{\pm0.7}$ | $21.3_{\pm7.0}$ | $\mathbf{24.3_{\pm1.8}}$ |
| | 50% | $12.0_{\pm0.0}$ | $22.3_{\pm6.3}$ | $\mathbf{28.7_{\pm1.4}}$ |
| | 70% | $14.3_{\pm0.3}$ | $\mathbf{33.7_{\pm2.2}}$ | $33.3_{\pm3.8}$ |
| | 90% | $16.0_{\pm0.6}$ | $14.3_{\pm2.9}$ | $\mathbf{19.0_{\pm6.1}}$ |

### D.4. Ablation studies for the speedups from OT

Carrying on from the previous section about the necessity of both the NeuralUCB and OT components, we can quantify through experiments the amount of speedups brought by OT. Specifically, we measure the wall clock time speedups for different sizes of the domain space $Q_t$ that we sample. As shown in Tab. 9, the larger the domain space $|Q_t|$, the greater the relative gain in the speedups. This mitigates the computational issues of applying NeuralUCB to the problem of exemplar selection and contributes positively to the success of EASE.

Table 9: Wall clock time speedups of using OT. The time in the table measures the wall clock time for each iteration of the algorithms.

| Domain size $|Q_t|$ | Time without OT (Sec) | Time with OT (Sec) | Speedup |
|---|---|---|---|
| 5000 | $21.48_{\pm0.12}$ | $5.62_{\pm0.05}$ | $3.8\times$ |
| 10000 | $44.16_{\pm0.67}$ | $6.92_{\pm0.02}$ | $6.4\times$ |
| 20000 | $91.39_{\pm1.06}$ | $8.82_{\pm0.12}$ | $10.4\times$ |
| 50000 | $216.61_{\pm1.25}$ | $15.32_{\pm0.04}$ | $14.1\times$ |

Table 10: Average validation accuracy $\pm$ standard error for $k = 50$ and $n = 1000$.

| | Evo | Best-of-N | EASE |
|---|---|---|---|
| AG News Remap | $50.0_{\pm 2.9}$ | $55.0_{\pm 2.9}$ | $\mathbf{68.3_{\pm 1.7}}$ |
| SST5 Reverse | $58.3_{\pm 1.7}$ | $56.7_{\pm 1.7}$ | $\mathbf{63.3_{\pm 1.7}}$ |

Table 11: Average test accuracy $\pm$ standard error for $k = 50$ and $n = 1000$.

| | Evo | Best-of-N | EASE |
|---|---|---|---|
| AG News Remap | $12.7_{\pm 3.2}$ | $26.3_{\pm 0.9}$ | $\mathbf{38.7_{\pm 3.2}}$ |
| SST5 Reverse | $39.0_{\pm 0.6}$ | $38.7_{\pm 1.2}$ | $\mathbf{40.0_{\pm 3.5}}$ |

Table 12: Average validation accuracy $\pm$ standard error for ablation studies of using an embedding without considering order.

| Task | Noise | AvgEmb | EASE |
|---|---|---|---|
| LR | 0% | $76.7_{\pm 1.7}$ | $\mathbf{80.0_{\pm 2.9}}$ |
| | 10% | $73.3_{\pm 4.4}$ | $\mathbf{73.3_{\pm 1.7}}$ |
| | 30% | $70.0_{\pm 0.0}$ | $\mathbf{76.7_{\pm 1.7}}$ |
| | 50% | $68.3_{\pm 4.4}$ | $\mathbf{78.3_{\pm 4.4}}$ |
| | 70% | $\mathbf{66.7_{\pm 1.7}}$ | $\mathbf{66.7_{\pm 1.7}}$ |
| | 90% | $51.7_{\pm 1.7}$ | $\mathbf{53.3_{\pm 1.7}}$ |
| LP-variant | 0% | $\mathbf{76.7_{\pm 1.7}}$ | $75.0_{\pm 0.0}$ |
| | 10% | $75.0_{\pm 0.0}$ | $\mathbf{80.0_{\pm 2.9}}$ |
| | 30% | $70.0_{\pm 0.0}$ | $\mathbf{75.0_{\pm 0.0}}$ |
| | 50% | $\mathbf{78.3_{\pm 1.7}}$ | $\mathbf{78.3_{\pm 1.7}}$ |
| | 70% | $\mathbf{75.0_{\pm 2.9}}$ | $71.7_{\pm 1.7}$ |
| | 90% | $61.7_{\pm 1.7}$ | $\mathbf{66.7_{\pm 1.7}}$ |
| AG News Remap | 0% | $\mathbf{53.3_{\pm 1.7}}$ | $50.0_{\pm 0.0}$ |
| | 10% | $50.0_{\pm 5.0}$ | $\mathbf{51.7_{\pm 1.7}}$ |
| | 30% | $51.7_{\pm 1.7}$ | $\mathbf{55.0_{\pm 0.0}}$ |
| | 50% | $50.0_{\pm 0.0}$ | $\mathbf{55.0_{\pm 2.9}}$ |
| | 70% | $\mathbf{58.3_{\pm 1.7}}$ | $58.3_{\pm 3.3}$ |
| | 90% | $53.3_{\pm 6.0}$ | $\mathbf{53.3_{\pm 1.7}}$ |
| SST5 Reverse | 0% | $46.7_{\pm 6.7}$ | $\mathbf{50.0_{\pm 2.9}}$ |
| | 10% | $\mathbf{50.0_{\pm 2.9}}$ | $48.3_{\pm 1.7}$ |
| | 30% | $45.0_{\pm 2.9}$ | $\mathbf{46.7_{\pm 3.3}}$ |
| | 50% | $\mathbf{48.3_{\pm 1.7}}$ | $46.7_{\pm 3.3}$ |
| | 70% | $41.7_{\pm 4.4}$ | $\mathbf{45.0_{\pm 5.0}}$ |
| | 90% | $\mathbf{31.7_{\pm 1.7}}$ | $\mathbf{31.7_{\pm 1.7}}$ |
| # best-performing tasks | | 9 | 18 |

Table 13: Average test accuracy $\pm$ standard error for ablation studies of using an embedding without considering order.

| Task | Noise | AvgEmb | EASE |
|---|---|---|---|
| LR | 0% | $48.7_{\pm 1.2}$ | $\mathbf{58.0_{\pm 2.1}}$ |
| | 10% | $\mathbf{48.7_{\pm 2.9}}$ | $42.3_{\pm 0.9}$ |
| | 30% | $\mathbf{53.0_{\pm 4.9}}$ | $47.7_{\pm 5.8}$ |
| | 50% | $46.0_{\pm 5.0}$ | $\mathbf{50.7_{\pm 2.2}}$ |
| | 70% | $44.3_{\pm 6.4}$ | $\mathbf{47.0_{\pm 3.2}}$ |
| | 90% | $28.7_{\pm 2.7}$ | $\mathbf{39.0_{\pm 7.8}}$ |
| LP-variant | 0% | $47.7_{\pm 1.9}$ | $\mathbf{53.0_{\pm 2.3}}$ |
| | 10% | $\mathbf{48.0_{\pm 1.7}}$ | $48.0_{\pm 2.5}$ |
| | 30% | $49.3_{\pm 1.8}$ | $\mathbf{50.7_{\pm 1.4}}$ |
| | 50% | $48.0_{\pm 1.5}$ | $\mathbf{56.7_{\pm 0.9}}$ |
| | 70% | $\mathbf{54.3_{\pm 1.3}}$ | $52.0_{\pm 1.0}$ |
| | 90% | $38.3_{\pm 0.9}$ | $\mathbf{44.3_{\pm 1.4}}$ |
| AG News Remap | 0% | $\mathbf{33.3_{\pm 2.3}}$ | $30.3_{\pm 1.2}$ |
| | 10% | $\mathbf{32.0_{\pm 0.6}}$ | $30.3_{\pm 4.4}$ |
| | 30% | $29.7_{\pm 1.2}$ | $\mathbf{40.0_{\pm 1.0}}$ |
| | 50% | $28.7_{\pm 1.4}$ | $\mathbf{40.0_{\pm 3.1}}$ |
| | 70% | $40.0_{\pm 5.0}$ | $\mathbf{45.0_{\pm 2.0}}$ |
| | 90% | $36.3_{\pm 4.1}$ | $\mathbf{36.7_{\pm 3.2}}$ |
| SST5 Reverse | 0% | $31.0_{\pm 4.6}$ | $\mathbf{34.0_{\pm 3.0}}$ |
| | 10% | $31.3_{\pm 3.0}$ | $\mathbf{31.3_{\pm 2.6}}$ |
| | 30% | $\mathbf{27.7_{\pm 1.4}}$ | $24.3_{\pm 1.8}$ |
| | 50% | $\mathbf{33.7_{\pm 1.4}}$ | $28.7_{\pm 1.4}$ |
| | 70% | $25.7_{\pm 7.2}$ | $\mathbf{33.3_{\pm 3.8}}$ |
| | 90% | $13.7_{\pm 0.3}$ | $\mathbf{19.0_{\pm 6.1}}$ |
| # best-performing tasks | | 8 | 16 |

## D.5. Ablation studies for other numbers of $k$

We validate that EASE is able to select a larger set of exemplars (i.e., larger $k$) and also from a larger exemplar set of size $n$. We conduct comparisons of EASE to the two most competitive baselines, Evo and Best-of-N, on the AG News Remap and SST5 Reverse datasets of size $n = 1000$ with 10% noise. According to Tab. 10 and Tab. 11, EASE performs much better than the two baselines. Therefore, EASE is able to work in other general setups with a larger $k$ and $n$.

We could potentially choose more exemplars to maximize the context window, but this will incur a very high future cost of inference (i.e., the cost of actually using the exemplars for inference during production). For example, to maximize the GPT-3.5-Turbo's context window of 16385 tokens (which OpenAI charges US\$0.5 per 1M tokens), each inference will cost US\$0.0081925 and each run of the algorithm costs US\$27.04 (assuming 165 iterations of query budget and 20 validation data samples). Similarly for GPT-4-Turbo with a context window of 128000 tokens (which OpenAI charges US\$10 per 1M tokens), each inference of the algorithm now costs US\$1.28 and each run of the algorithm costs US\$4224. Therefore, while it is theoretically possible to use even larger $k$ which exhausts the context window, it is not economically viable to conduct experiments or adopt such an approach in practice.

### D.6. Ablation studies for the benefit of having an ordering-aware embedding in `EASE`

Note that even for the same subset of exemplars, a different ordering leads to different pre-trained embedding from embedding model $h(\cdot)$. In this section, we investigate the impact of capturing the ordering of the exemplars in the exemplar sequence $E$. To contrast the adopted approach, we propose a new embedding that simply averages the embeddings of all exemplars in the chosen subset. So, this embedding will be invariant with regard to the ordering of exemplars and we call it AvgEmb. As demonstrated in Tab. 12 and Tab. 13, adopting an ordering-aware embedding in `EASE` results in better overall performance as compared to AvgEmd which disregards exemplar ordering.

### D.7. Ablation studies for optimizing exemplars for different black-box LLMs in `EASE`

We perform exemplar selection for other target black-box models that are not GPT-3.5 which we used for all other experiments previously. We use GPT-4-V (i.e., "gpt-4-turbo-2024-04-09" with vision capability), GPT-4-Turbo (i.e., 'gpt-4-1106-preview' without vision capability) and Gemini Pro (Team et al., 2024) (i.e., "gemini-1.0-pro") for our experiments. Tab. 14 and Tab. 15 show that our `EASE` is able to select effective exemplars for different black-box models. Note that the performance of GPT-4-V and GPT-4-Turbo is comparable to or worse than that of GPT-3.5 in some tasks. This may be attributed to significant variations in performance across different versions (i.e., checkpoints) of the GPT models. For example, existing work by Chen et al. (2023b) has shown that GPT-4's mathematical ability varies a lot across different checkpoints, with some exhibiting poor performance on mathematical problems. This variability partially explains the suboptimal performance of GPT-4-V and GPT-4-Turbo on the task of LR. Additionally, it is worth investigating, in future work, the significance of exemplar selection as the LLMs continue to become increasingly powerful.

Table 14: Average validation accuracy $\pm$ standard error when using our `EASE` to select exemplars for different target black-box models.

| Task (with 10% noise) | GPT-4-V | GPT-4-Turbo | Gemini Pro |
|---|---|---|---|
| LR | $1.7_{\pm 1.7}$ | $3.3_{\pm 1.7}$ | $83.3_{\pm 4.4}$ |
| LP-variant | $90.0_{\pm 0.0}$ | $90.0_{\pm 0.0}$ | $31.7_{\pm 1.7}$ |
| AG News Remap | $50.0_{\pm 0.0}$ | $35.0_{\pm 2.9}$ | $53.3_{\pm 4.4}$ |
| SST5 Reverse | $21.7_{\pm 1.7}$ | $41.7_{\pm 1.7}$ | $36.7_{\pm 1.7}$ |

Table 15: Average test accuracy $\pm$ standard error when using our `EASE` to select exemplars for different target black-box models.

| Task (with 10% noise) | GPT-4-V | GPT-4-Turbo | Gemini Pro |
|---|---|---|---|
| LR | $0.0_{\pm 0.0}$ | $0.0_{\pm 0.0}$ | $51.3_{\pm 3.5}$ |
| LP-variant | $83.7_{\pm 1.3}$ | $77.7_{\pm 1.9}$ | $14.3_{\pm 1.4}$ |
| AG News Remap | $25.7_{\pm 1.9}$ | $25.7_{\pm 3.0}$ | $32.7_{\pm 9.4}$ |
| SST5 Reverse | $13.3_{\pm 1.4}$ | $28.3_{\pm 3.8}$ | $24.0_{\pm 3.0}$ |

### D.8. Ablation studies for using different embedding models in `EASE`

For our main experiments, we use MPNet as the embedding model for our `EASE`. Here, we use MiniLM (Wang et al., 2020) and CLIP (Radford et al., 2021) to obtain the embedding for our `EASE`, respectively. Tab. 16 and Tab. 17 show that our `EASE` achieves competitive performance using different embedding models. Therefore, our `EASE` is general in the sense that different embedding models can be used.

Table 16: Average validation accuracy $\pm$ standard error when using different embedding models for our `EASE`.

| Task (with 10% noise) | MiniLM | CLIP |
|---|---|---|
| LR | $73.3_{\pm 1.7}$ | $68.3_{\pm 1.7}$ |
| LP-variant | $76.7_{\pm 1.7}$ | $70.0_{\pm 2.9}$ |
| AG News Remap | $43.3_{\pm 1.7}$ | $46.7_{\pm 3.3}$ |
| SST5 Reverse | $43.3_{\pm 1.7}$ | $45.0_{\pm 0.0}$ |

Table 17: Average test accuracy $\pm$ standard error when using different embedding models for our `EASE`.

| Task (with 10% noise) | MiniLM | CLIP |
|---|---|---|
| LR | $51.3_{\pm 1.3}$ | $50.3_{\pm 5.5}$ |
| LP-variant | $56.3_{\pm 0.9}$ | $49.7_{\pm 1.4}$ |
| AG News Remap | $29.3_{\pm 2.7}$ | $29.3_{\pm 0.3}$ |
| SST5 Reverse | $29.7_{\pm 6.4}$ | $33.3_{\pm 4.8}$ |

### D.9. Ablation studies for the degree of exploration

One important hyperparameter of the NeuralUCB algorithm utilized in the paper is $\nu$ (see (3)), which controls the degree of exploration performed during the optimization process. Our finding indicates that a small $\nu$ in `EASE` results in better exemplar selection performance. As demonstrated in Tab. 18 and Tab. 19, having a near-zero (i.e., $\nu = 0.01$) exploration is the best. This may be attributed to our sub-sampling of the domain space $\Omega$, which is motivated by practically reducing

computational complexities, and at the same time inherently serves as a form of exploration. Given the enormous size of the $\Omega$, sub-sampling to $Q_t$ likely generates a different exemplar sequences space within each iteration of the algorithm. This observation is further supported by Bastani et al. (2021), who found that a greedy algorithm can be rate optimal given sufficient randomness in the observed contexts of the bandits algorithm. Therefore, only a small extent of exploration is required for EASE.

To further support the above observation empirically, we conduct additional experiments with the domain space $Q_t$ fixed throughout. When $|Q_t| = 1000$ and fixed, we observe results in Tab. 20 and Tab. 21 that a larger value of $\nu$ performs better. Thus, exploration is essential when there is no randomness in the context (i.e., the domain space $Q_t$). However, given sufficient randomness in our EASE algorithm from sub-sampling, we adopt a small $\nu = 0.01$ for optimal performance.

Table 18: Average accuracy $\pm$ standard error for ablation studies of the exploration parameter $\nu$.

| Task | Noise | $\nu = 0$ | $\nu = 0.01$ | $\nu = 0.1$ | $\nu = 1$ |
|---|---|---|---|---|---|
| LP-variant | 0% | **75.0**$_{\pm 0.0}$ | **75.0**$_{\pm 0.0}$ | 71.7$_{\pm 1.7}$ | 70.0$_{\pm 0.0}$ |
| | 10% | 75.0$_{\pm 2.9}$ | **80.0**$_{\pm 2.9}$ | 71.7$_{\pm 1.7}$ | 73.3$_{\pm 1.7}$ |
| | 30% | 73.3$_{\pm 1.7}$ | **75.0**$_{\pm 0.0}$ | 68.3$_{\pm 1.7}$ | 71.7$_{\pm 1.7}$ |
| | 50% | 76.7$_{\pm 3.3}$ | **78.3**$_{\pm 1.7}$ | 75.0$_{\pm 2.9}$ | 71.7$_{\pm 1.7}$ |
| | 70% | **75.0**$_{\pm 0.0}$ | 71.7$_{\pm 1.7}$ | 73.3$_{\pm 1.7}$ | 70.0$_{\pm 0.0}$ |
| | 90% | 63.3$_{\pm 1.7}$ | **66.7**$_{\pm 1.7}$ | 63.3$_{\pm 1.7}$ | 63.3$_{\pm 1.7}$ |

Table 19: Average test accuracy $\pm$ standard error for ablation studies of the exploration parameter $\nu$.

| Task | Noise | $\nu = 0$ | $\nu = 0.01$ | $\nu = 0.1$ | $\nu = 1$ |
|---|---|---|---|---|---|
| LP-variant | 0% | 49.3$_{\pm 2.4}$ | **53.0**$_{\pm 2.3}$ | 52.0$_{\pm 1.1}$ | 50.0$_{\pm 3.5}$ |
| | 10% | 46.0$_{\pm 1.0}$ | 48.0$_{\pm 2.5}$ | **51.3**$_{\pm 2.2}$ | 49.3$_{\pm 1.9}$ |
| | 30% | 49.3$_{\pm 1.3}$ | 50.7$_{\pm 1.4}$ | 43.7$_{\pm 3.8}$ | **52.3**$_{\pm 2.4}$ |
| | 50% | 54.3$_{\pm 2.6}$ | **56.7**$_{\pm 0.9}$ | 50.3$_{\pm 2.7}$ | 47.0$_{\pm 1.1}$ |
| | 70% | 49.0$_{\pm 4.0}$ | **52.0**$_{\pm 1.0}$ | 46.0$_{\pm 2.5}$ | 47.0$_{\pm 1.1}$ |
| | 90% | 41.3$_{\pm 2.3}$ | **44.3**$_{\pm 1.4}$ | 36.7$_{\pm 2.6}$ | 40.3$_{\pm 1.4}$ |

Table 20: Average accuracy $\pm$ standard error for ablation studies of exploration in a fixed domain.

| Task | Noise | $\nu = 0$ | $\nu = 0.01$ | $\nu = 0.1$ | $\nu = 1$ |
|---|---|---|---|---|---|
| LP-variant | 0% | 60.0$_{\pm 0.0}$ | 66.7$_{\pm 1.7}$ | **68.3**$_{\pm 1.7}$ | 66.7$_{\pm 1.7}$ |
| | 10% | 60.0$_{\pm 0.0}$ | 61.7$_{\pm 1.7}$ | **66.7**$_{\pm 1.7}$ | 65.0$_{\pm 2.9}$ |
| | 30% | 58.3$_{\pm 1.7}$ | 58.3$_{\pm 1.7}$ | **65.0**$_{\pm 2.9}$ | **65.0**$_{\pm 2.9}$ |
| | 50% | 60.0$_{\pm 2.9}$ | 65.0$_{\pm 0.0}$ | 61.7$_{\pm 1.7}$ | **66.7**$_{\pm 1.7}$ |
| | 70% | 50.0$_{\pm 0.0}$ | 51.7$_{\pm 1.7}$ | 58.3$_{\pm 1.7}$ | **61.7**$_{\pm 3.3}$ |
| | 90% | 36.7$_{\pm 1.7}$ | 36.7$_{\pm 1.7}$ | 45.0$_{\pm 0.0}$ | **48.3**$_{\pm 1.7}$ |

Table 21: Average test accuracy $\pm$ standard error for ablation studies of the exploration in a fixed domain.

| Task | Noise | $\nu = 0$ | $\nu = 0.01$ | $\nu = 0.1$ | $\nu = 1$ |
|---|---|---|---|---|---|
| LP-variant | 0% | 38.3$_{\pm 0.9}$ | 51.0$_{\pm 3.2}$ | **53.7**$_{\pm 3.3}$ | 45.0$_{\pm 1.1}$ |
| | 10% | 41.7$_{\pm 2.7}$ | 42.7$_{\pm 1.9}$ | 47.0$_{\pm 1.5}$ | **49.0**$_{\pm 2.1}$ |
| | 30% | 39.3$_{\pm 2.3}$ | 40.3$_{\pm 3.3}$ | **43.3**$_{\pm 1.8}$ | 43.0$_{\pm 1.1}$ |
| | 50% | 43.3$_{\pm 2.3}$ | 47.7$_{\pm 0.9}$ | 41.7$_{\pm 2.3}$ | **48.7**$_{\pm 2.4}$ |
| | 70% | 39.7$_{\pm 1.4}$ | 42.0$_{\pm 3.1}$ | 39.7$_{\pm 1.9}$ | **48.7**$_{\pm 0.3}$ |
| | 90% | 20.7$_{\pm 0.3}$ | 22.3$_{\pm 0.9}$ | 29.0$_{\pm 3.6}$ | **29.0**$_{\pm 1.1}$ |

### D.10. Ablation studies: Asking GPT to directly select exemplars

One may wonder whether ChatGPT can directly help us select the exemplars for in-context learning. To test the possibility of directly utilizing ChatGPT, we use the following prompt to query the GPT:

> *"You are asked to perform a task that is described by the examples below, the goal is to correctly give output based on the input. Given the numbered list of examples below, pick 5 of them that will best serve as examples for in-context learning for this task. Only output the list of numbers."*

We provide all the exemplars in $D$ in the form of a numbered list to GPT together with the prompt above, and obtain the GPT-selected exemplars. We call this method GPT Select. As shown in Tab. 22 and Tab. 23, the exemplars directly selected by GPT perform worse than EASE for ICL.

### D.11. Test accuracies for all tables

We present the test accuracies for all tables presented in the main paper. Note that we report validation accuracies in the main tables because the effectiveness of the optimization strategy is directly reflected by the maximized value of the objective function in (1) at the end of all iterations. Nevertheless, we show all test accuracies here for reference and completeness. The test accuracy can be affected by the difference in the distribution of the validation and test data, which is out of our control. This difference in distribution is significant in our case because the validation set only contains 20 data points (limited by the

Table 22: Average validation accuracy $\pm$ standard error when asking GPT to directly output the best exemplars.

| Task (with 10% noise) | GPT Select | EASE |
|---|---|---|
| LR | $35.0_{\pm7.6}$ | $\mathbf{73.3_{\pm1.7}}$ |
| LP-variant | $58.3_{\pm3.3}$ | $\mathbf{80.0_{\pm2.9}}$ |
| AG News Remap | $10.0_{\pm0.0}$ | $\mathbf{51.7_{\pm1.7}}$ |
| SST5 Reverse | $10.0_{\pm0.0}$ | $\mathbf{48.3_{\pm1.7}}$ |

Table 23: Average test accuracy $\pm$ standard error when asking GPT to directly output the best exemplars.

| Task (with 10% noise) | GPT Select | EASE |
|---|---|---|
| LR | $29.0_{\pm9.0}$ | $\mathbf{42.3_{\pm0.9}}$ |
| LP-variant | $44.0_{\pm1.0}$ | $\mathbf{48.0_{\pm2.5}}$ |
| AG News Remap | $12.7_{\pm0.3}$ | $\mathbf{30.3_{\pm4.4}}$ |
| SST5 Reverse | $10.0_{\pm0.0}$ | $\mathbf{31.3_{\pm2.6}}$ |

Table 24: Test accuracies counterpart of Tab. 1 over 3 independent trials.

| | DPP | MMD | OT | Cosine | BM25 | Active | Inf | Evo | Best-of-N | EASE |
|---|---|---|---|---|---|---|---|---|---|---|
| active_to_passive | $\mathbf{100.0_{\pm0.0}}$ | $\mathbf{100.0_{\pm0.0}}$ | $\mathbf{100.0_{\pm0.0}}$ | $\mathbf{100.0_{\pm0.0}}$ | $\mathbf{100.0_{\pm0.0}}$ | $\mathbf{100.0_{\pm0.0}}$ | $\mathbf{100.0_{\pm0.0}}$ | $\mathbf{100.0_{\pm0.0}}$ | $\mathbf{100.0_{\pm0.0}}$ | $\mathbf{100.0_{\pm0.0}}$ |
| antonyms | $75.7_{\pm0.3}$ | $\mathbf{86.0_{\pm0.0}}$ | $85.3_{\pm0.7}$ | $77.7_{\pm0.3}$ | $80.7_{\pm0.3}$ | $80.0_{\pm2.5}$ | $82.3_{\pm0.3}$ | $82.0_{\pm0.0}$ | $84.3_{\pm0.3}$ | $84.7_{\pm0.3}$ |
| auto_categorization | $28.7_{\pm0.3}$ | $25.7_{\pm0.7}$ | $24.0_{\pm0.6}$ | $\mathbf{39.3_{\pm0.9}}$ | $27.0_{\pm1.0}$ | $32.7_{\pm1.0}$ | $35.0_{\pm1.5}$ | $17.3_{\pm0.3}$ | $30.7_{\pm0.7}$ | $34.0_{\pm2.1}$ |
| diff | $2.0_{\pm0.0}$ | $2.0_{\pm0.0}$ | $2.0_{\pm0.0}$ | $2.0_{\pm0.0}$ | $2.0_{\pm0.0}$ | $\mathbf{100.0_{\pm0.0}}$ | $\mathbf{100.0_{\pm0.0}}$ | $\mathbf{100.0_{\pm0.0}}$ | $\mathbf{100.0_{\pm0.0}}$ | $\mathbf{100.0_{\pm0.0}}$ |
| first_word_letter | $\mathbf{100.0_{\pm0.0}}$ | $\mathbf{100.0_{\pm0.0}}$ | $\mathbf{100.0_{\pm0.0}}$ | $\mathbf{100.0_{\pm0.0}}$ | $\mathbf{100.0_{\pm0.0}}$ | $\mathbf{100.0_{\pm0.0}}$ | $\mathbf{100.0_{\pm0.0}}$ | $\mathbf{100.0_{\pm0.0}}$ | $\mathbf{100.0_{\pm0.0}}$ | $\mathbf{100.0_{\pm0.0}}$ |
| larger_animal | $69.0_{\pm0.6}$ | $78.7_{\pm1.2}$ | $83.7_{\pm0.7}$ | $77.7_{\pm0.7}$ | $84.0_{\pm1.0}$ | $61.0_{\pm1.7}$ | $84.7_{\pm0.3}$ | $\mathbf{90.0_{\pm0.6}}$ | $87.7_{\pm2.9}$ | $88.0_{\pm0.0}$ |
| letters_list | $\mathbf{100.0_{\pm0.0}}$ | $\mathbf{100.0_{\pm0.0}}$ | $\mathbf{100.0_{\pm0.0}}$ | $\mathbf{100.0_{\pm0.0}}$ | $\mathbf{100.0_{\pm0.0}}$ | $\mathbf{100.0_{\pm0.0}}$ | $\mathbf{100.0_{\pm0.0}}$ | $\mathbf{100.0_{\pm0.0}}$ | $\mathbf{100.0_{\pm0.0}}$ | $\mathbf{100.0_{\pm0.0}}$ |
| negation | $85.7_{\pm0.7}$ | $87.3_{\pm0.7}$ | $86.3_{\pm0.3}$ | $87.7_{\pm0.3}$ | $83.3_{\pm0.3}$ | $85.7_{\pm1.1}$ | $83.7_{\pm0.3}$ | $86.3_{\pm0.3}$ | $85.0_{\pm0.6}$ | $\mathbf{89.0_{\pm0.6}}$ |
| num_to_verbal | $97.3_{\pm0.3}$ | $98.7_{\pm0.3}$ | $\mathbf{99.0_{\pm0.0}}$ | $\mathbf{99.0_{\pm0.0}}$ | $96.0_{\pm0.0}$ | $97.7_{\pm0.7}$ | $98.0_{\pm0.0}$ | $96.7_{\pm0.3}$ | $96.7_{\pm0.3}$ | $96.3_{\pm0.3}$ |
| object_counting | $45.3_{\pm0.9}$ | $48.3_{\pm0.9}$ | $40.7_{\pm0.3}$ | $27.0_{\pm2.5}$ | $31.3_{\pm3.8}$ | $48.0_{\pm2.0}$ | $\mathbf{58.0_{\pm0.6}}$ | $42.3_{\pm0.3}$ | $52.3_{\pm3.3}$ | $52.0_{\pm2.1}$ |
| orthography_starts_with | $30.3_{\pm0.9}$ | $42.0_{\pm0.6}$ | $65.7_{\pm0.3}$ | $65.7_{\pm0.9}$ | $\mathbf{71.0_{\pm0.6}}$ | $40.3_{\pm7.8}$ | $69.3_{\pm0.9}$ | $47.0_{\pm1.0}$ | $63.3_{\pm2.4}$ | $69.3_{\pm0.7}$ |
| rhymes | $58.0_{\pm1.0}$ | $42.0_{\pm0.6}$ | $11.3_{\pm0.9}$ | $\mathbf{99.0_{\pm0.0}}$ | $64.3_{\pm1.9}$ | $47.3_{\pm6.9}$ | $59.3_{\pm15.4}$ | $98.3_{\pm0.3}$ | $96.0_{\pm0.0}$ | $97.7_{\pm1.3}$ |
| second_word_letter | $17.3_{\pm0.9}$ | $31.0_{\pm0.0}$ | $32.3_{\pm0.7}$ | $42.0_{\pm3.1}$ | $42.3_{\pm0.9}$ | $27.0_{\pm8.2}$ | $45.7_{\pm1.2}$ | $40.3_{\pm0.3}$ | $38.7_{\pm0.3}$ | $\mathbf{48.3_{\pm1.4}}$ |
| sentence_similarity | $19.0_{\pm0.6}$ | $13.7_{\pm0.3}$ | $38.3_{\pm1.4}$ | $\mathbf{41.0_{\pm0.6}}$ | $33.7_{\pm2.3}$ | $19.0_{\pm1.7}$ | $20.0_{\pm5.0}$ | $18.0_{\pm0.0}$ | $31.3_{\pm2.7}$ | $32.7_{\pm1.2}$ |
| sentiment | $91.3_{\pm0.3}$ | $89.0_{\pm0.0}$ | $87.0_{\pm0.0}$ | $90.3_{\pm0.3}$ | $90.7_{\pm0.7}$ | $\mathbf{91.7_{\pm0.7}}$ | $89.0_{\pm0.0}$ | $89.7_{\pm0.3}$ | $89.3_{\pm0.3}$ | $89.0_{\pm0.0}$ |
| singular_to_plural | $\mathbf{100.0_{\pm0.0}}$ | $\mathbf{100.0_{\pm0.0}}$ | $\mathbf{100.0_{\pm0.0}}$ | $\mathbf{100.0_{\pm0.0}}$ | $\mathbf{100.0_{\pm0.0}}$ | $\mathbf{100.0_{\pm0.0}}$ | $\mathbf{100.0_{\pm0.0}}$ | $\mathbf{100.0_{\pm0.0}}$ | $\mathbf{100.0_{\pm0.0}}$ | $\mathbf{100.0_{\pm0.0}}$ |
| sum | $2.0_{\pm0.0}$ | $2.0_{\pm0.0}$ | $2.0_{\pm0.0}$ | $2.0_{\pm0.0}$ | $2.0_{\pm0.0}$ | $34.7_{\pm26.7}$ | $\mathbf{100.0_{\pm0.0}}$ | $\mathbf{100.0_{\pm0.0}}$ | $\mathbf{100.0_{\pm0.0}}$ | $\mathbf{100.0_{\pm0.0}}$ |
| synonyms | $14.0_{\pm0.6}$ | $14.3_{\pm0.7}$ | $11.0_{\pm0.0}$ | $14.0_{\pm0.0}$ | $13.7_{\pm0.3}$ | $13.0_{\pm0.5}$ | $14.7_{\pm0.9}$ | $13.3_{\pm0.9}$ | $\mathbf{18.3_{\pm0.3}}$ | $13.7_{\pm0.3}$ |
| taxonomy_animal | $47.7_{\pm0.3}$ | $44.7_{\pm1.2}$ | $77.3_{\pm1.9}$ | $73.3_{\pm0.3}$ | $67.3_{\pm5.9}$ | $46.3_{\pm7.1}$ | $62.3_{\pm8.9}$ | $32.7_{\pm0.7}$ | $73.7_{\pm4.2}$ | $\mathbf{78.7_{\pm1.9}}$ |
| translation_en-de | $83.7_{\pm0.3}$ | $84.3_{\pm0.3}$ | $83.3_{\pm0.3}$ | $83.0_{\pm0.6}$ | $83.3_{\pm1.3}$ | $80.3_{\pm1.9}$ | $\mathbf{85.0_{\pm1.1}}$ | $83.3_{\pm0.3}$ | $82.3_{\pm0.3}$ | $83.3_{\pm0.3}$ |
| translation_en-es | $83.7_{\pm0.3}$ | $\mathbf{89.7_{\pm0.3}}$ | $88.3_{\pm0.3}$ | $87.3_{\pm1.2}$ | $87.3_{\pm1.2}$ | $88.0_{\pm0.8}$ | $86.7_{\pm1.4}$ | $84.3_{\pm0.3}$ | $84.3_{\pm0.3}$ | $84.7_{\pm0.9}$ |
| translation_en-fr | $83.0_{\pm0.0}$ | $87.7_{\pm0.3}$ | $87.3_{\pm0.3}$ | $87.0_{\pm0.0}$ | $87.0_{\pm0.0}$ | $83.3_{\pm1.7}$ | $87.7_{\pm0.7}$ | $86.7_{\pm0.3}$ | $\mathbf{88.3_{\pm0.3}}$ | $85.0_{\pm1.0}$ |
| word_sorting | $8.7_{\pm0.7}$ | $65.0_{\pm0.6}$ | $66.7_{\pm1.2}$ | $66.0_{\pm1.1}$ | $43.7_{\pm2.3}$ | $\mathbf{71.0_{\pm1.4}}$ | $69.3_{\pm1.7}$ | $61.3_{\pm1.2}$ | $63.3_{\pm0.7}$ | $69.7_{\pm0.3}$ |
| word_unscrambling | $61.3_{\pm0.3}$ | $58.7_{\pm0.7}$ | $\mathbf{63.3_{\pm0.3}}$ | $57.0_{\pm1.1}$ | $61.3_{\pm0.7}$ | $60.7_{\pm1.7}$ | $58.3_{\pm1.4}$ | $53.7_{\pm0.3}$ | $60.3_{\pm0.3}$ | $62.0_{\pm2.1}$ |
| # best-performing tasks | 4 | 6 | 6 | 8 | 5 | 7 | 8 | 7 | 8 | 9 |

fact that querying the GPT-3.5 API is expensive). This setup is disadvantageous for EASE because the optimized exemplar is "overfitted" to the validation set. This test performance is expected to improve much if we use a larger validation set (e.g., with 100 validation samples).

Table 25: Test accuracies counterpart of Tab. 2 over 3 independent trials.

| Type | Task | Noise | DPP | MMD | OT | Cosine | BM25 | Active | Inf | Evo | Best-of-N | EASE |
|---|---|---|---|---|---|---|---|---|---|---|---|---|
| Rule-based tasks | LR | 0% | $36.7_{\pm0.7}$ | $38.0_{\pm1.0}$ | $34.0_{\pm1.0}$ | $48.0_{\pm1.1}$ | $40.7_{\pm2.3}$ | $32.0_{\pm6.6}$ | $43.3_{\pm5.8}$ | $38.0_{\pm0.6}$ | $47.7_{\pm0.3}$ | $\mathbf{58.0_{\pm2.1}}$ |
| | | 10% | $7.3_{\pm0.9}$ | $33.3_{\pm0.7}$ | $29.3_{\pm0.3}$ | $40.3_{\pm3.8}$ | $41.0_{\pm2.9}$ | $0.0_{\pm0.0}$ | $46.3_{\pm1.2}$ | $38.3_{\pm0.3}$ | $\mathbf{47.7_{\pm0.7}}$ | $42.3_{\pm0.9}$ |
| | | 30% | $12.0_{\pm0.6}$ | $23.3_{\pm0.3}$ | $30.0_{\pm1.0}$ | $45.7_{\pm4.8}$ | $29.7_{\pm2.9}$ | $9.0_{\pm5.8}$ | $35.0_{\pm4.7}$ | $38.3_{\pm1.2}$ | $43.0_{\pm0.6}$ | $\mathbf{47.7_{\pm5.8}}$ |
| | | 50% | $3.0_{\pm1.0}$ | $33.7_{\pm0.3}$ | $23.0_{\pm0.6}$ | $43.7_{\pm2.3}$ | $33.3_{\pm0.7}$ | $0.0_{\pm0.0}$ | $47.0_{\pm2.1}$ | $32.7_{\pm0.9}$ | $30.3_{\pm1.8}$ | $\mathbf{50.7_{\pm2.2}}$ |
| | | 70% | $0.0_{\pm0.0}$ | $38.7_{\pm0.3}$ | $28.7_{\pm0.7}$ | $44.0_{\pm3.5}$ | $34.0_{\pm1.1}$ | $1.7_{\pm1.4}$ | $31.7_{\pm3.8}$ | $30.3_{\pm0.7}$ | $26.3_{\pm5.3}$ | $\mathbf{47.0_{\pm3.2}}$ |
| | | 90% | $0.0_{\pm0.0}$ | $32.7_{\pm1.3}$ | $32.0_{\pm0.6}$ | $37.3_{\pm4.8}$ | $2.3_{\pm0.9}$ | $0.0_{\pm0.0}$ | $7.7_{\pm1.9}$ | $5.0_{\pm0.6}$ | $14.0_{\pm0.6}$ | $\mathbf{39.0_{\pm7.8}}$ |
| | LP-variant | 0% | $39.3_{\pm0.7}$ | $35.0_{\pm0.6}$ | $34.7_{\pm0.7}$ | $47.3_{\pm0.3}$ | $36.3_{\pm1.4}$ | $34.0_{\pm3.7}$ | $\mathbf{53.3_{\pm1.3}}$ | $38.3_{\pm0.9}$ | $51.7_{\pm3.3}$ | $53.0_{\pm2.3}$ |
| | | 10% | $0.0_{\pm0.0}$ | $37.0_{\pm0.6}$ | $34.0_{\pm0.6}$ | $49.0_{\pm1.1}$ | $39.3_{\pm2.2}$ | $39.0_{\pm0.8}$ | $\mathbf{52.3_{\pm2.6}}$ | $37.7_{\pm0.7}$ | $48.0_{\pm1.1}$ | $48.0_{\pm2.5}$ |
| | | 30% | $0.0_{\pm0.0}$ | $44.3_{\pm0.7}$ | $36.0_{\pm1.1}$ | $38.7_{\pm4.3}$ | $34.3_{\pm1.9}$ | $36.7_{\pm2.4}$ | $48.0_{\pm3.1}$ | $35.7_{\pm0.3}$ | $49.3_{\pm0.7}$ | $\mathbf{50.7_{\pm1.4}}$ |
| | | 50% | $0.0_{\pm0.0}$ | $53.3_{\pm1.2}$ | $31.0_{\pm0.6}$ | $47.0_{\pm0.6}$ | $34.3_{\pm1.3}$ | $12.7_{\pm5.2}$ | $42.7_{\pm2.0}$ | $34.0_{\pm0.6}$ | $49.3_{\pm0.3}$ | $\mathbf{56.7_{\pm0.9}}$ |
| | | 70% | $0.0_{\pm0.0}$ | $39.0_{\pm1.5}$ | $30.3_{\pm0.9}$ | $46.7_{\pm1.3}$ | $36.3_{\pm0.7}$ | $4.3_{\pm3.5}$ | $51.0_{\pm1.7}$ | $33.3_{\pm0.7}$ | $31.3_{\pm0.9}$ | $\mathbf{52.0_{\pm1.0}}$ |
| | | 90% | $0.0_{\pm0.0}$ | $35.0_{\pm0.6}$ | $39.7_{\pm0.3}$ | $35.3_{\pm1.9}$ | $0.0_{\pm0.0}$ | $0.0_{\pm0.0}$ | $41.0_{\pm1.7}$ | $16.7_{\pm0.9}$ | $28.0_{\pm0.6}$ | $\mathbf{44.3_{\pm1.4}}$ |
| Re-mapped label tasks | AG News Remap | 0% | $7.0_{\pm0.6}$ | $7.0_{\pm0.0}$ | $12.3_{\pm0.7}$ | $30.3_{\pm1.4}$ | $28.3_{\pm1.3}$ | $4.3_{\pm1.1}$ | $9.0_{\pm2.5}$ | $11.3_{\pm0.3}$ | $19.3_{\pm0.7}$ | $\mathbf{30.3_{\pm1.2}}$ |
| | | 10% | $2.0_{\pm0.0}$ | $7.0_{\pm0.0}$ | $13.0_{\pm1.1}$ | $17.0_{\pm4.0}$ | $19.0_{\pm4.5}$ | $6.0_{\pm0.8}$ | $13.0_{\pm3.2}$ | $12.0_{\pm0.6}$ | $30.0_{\pm0.6}$ | $\mathbf{30.3_{\pm4.4}}$ |
| | | 30% | $3.7_{\pm0.3}$ | $1.0_{\pm0.0}$ | $4.7_{\pm0.3}$ | $28.0_{\pm2.0}$ | $22.3_{\pm0.7}$ | $7.7_{\pm2.2}$ | $5.0_{\pm0.6}$ | $12.3_{\pm0.3}$ | $28.0_{\pm5.0}$ | $\mathbf{40.0_{\pm1.0}}$ |
| | | 50% | $4.7_{\pm0.3}$ | $3.0_{\pm0.0}$ | $3.7_{\pm0.3}$ | $27.3_{\pm2.3}$ | $21.3_{\pm2.7}$ | $8.7_{\pm3.1}$ | $13.0_{\pm3.5}$ | $6.7_{\pm0.3}$ | $21.7_{\pm2.3}$ | $\mathbf{40.0_{\pm3.1}}$ |
| | | 70% | $2.0_{\pm0.0}$ | $23.3_{\pm0.3}$ | $7.0_{\pm0.0}$ | $20.0_{\pm1.0}$ | $16.0_{\pm0.6}$ | $15.7_{\pm7.1}$ | $5.0_{\pm1.1}$ | $4.7_{\pm0.3}$ | $36.0_{\pm0.6}$ | $\mathbf{45.0_{\pm2.0}}$ |
| | | 90% | $8.0_{\pm0.2}$ | $21.0_{\pm1.1}$ | $2.0_{\pm0.0}$ | $22.7_{\pm7.0}$ | $2.0_{\pm0.0}$ | $11.7_{\pm1.9}$ | $23.0_{\pm1.5}$ | $6.3_{\pm0.3}$ | $27.3_{\pm1.9}$ | $\mathbf{36.7_{\pm3.2}}$ |
| | SST5 Reverse | 0% | $13.3_{\pm0.7}$ | $11.7_{\pm0.7}$ | $9.7_{\pm0.9}$ | $22.0_{\pm3.6}$ | $18.3_{\pm0.7}$ | $10.3_{\pm0.3}$ | $21.7_{\pm5.5}$ | $11.3_{\pm0.3}$ | $23.7_{\pm0.7}$ | $\mathbf{34.0_{\pm3.0}}$ |
| | | 10% | $13.3_{\pm0.9}$ | $11.3_{\pm0.3}$ | $9.3_{\pm0.3}$ | $20.0_{\pm2.6}$ | $18.0_{\pm1.0}$ | $11.7_{\pm1.5}$ | $27.0_{\pm1.1}$ | $12.0_{\pm0.6}$ | $23.0_{\pm0.6}$ | $\mathbf{31.3_{\pm2.6}}$ |
| | | 30% | $15.7_{\pm0.3}$ | $13.3_{\pm0.3}$ | $11.7_{\pm0.7}$ | $19.3_{\pm2.3}$ | $23.7_{\pm0.7}$ | $9.7_{\pm0.3}$ | $21.3_{\pm2.7}$ | $11.0_{\pm0.6}$ | $13.0_{\pm1.5}$ | $\mathbf{24.3_{\pm1.8}}$ |
| | | 50% | $13.0_{\pm0.6}$ | $12.3_{\pm0.3}$ | $12.0_{\pm0.0}$ | $28.3_{\pm2.9}$ | $14.0_{\pm0.6}$ | $12.7_{\pm0.7}$ | $14.3_{\pm0.9}$ | $9.3_{\pm0.3}$ | $14.0_{\pm2.0}$ | $\mathbf{28.7_{\pm1.4}}$ |
| | | 70% | $12.0_{\pm0.0}$ | $12.0_{\pm0.6}$ | $14.3_{\pm0.3}$ | $24.7_{\pm0.7}$ | $12.0_{\pm1.1}$ | $11.7_{\pm0.5}$ | $18.7_{\pm2.9}$ | $15.7_{\pm0.3}$ | $33.0_{\pm1.0}$ | $\mathbf{33.3_{\pm3.8}}$ |
| | | 90% | $16.0_{\pm0.6}$ | $9.3_{\pm0.7}$ | $16.0_{\pm0.6}$ | $12.7_{\pm0.3}$ | $13.3_{\pm0.9}$ | $12.7_{\pm0.3}$ | $14.0_{\pm0.6}$ | $11.7_{\pm0.7}$ | $10.7_{\pm0.3}$ | $\mathbf{19.0_{\pm6.1}}$ |
| # best-performing tasks | | | 0 | 0 | 0 | 0 | 0 | 0 | 2 | 0 | 1 | 21 |

Table 26: Test accuracies counterpart of Tab. 3 over 3 independent trials.

| | EASE | EASE with instructions | improvement |
|---|---|---|---|
| antonyms | $\mathbf{84.7_{\pm0.3}}$ | $82.0_{\pm0.6}$ | -2.7 ↓ |
| auto_categorization | $34.0_{\pm2.1}$ | $\mathbf{48.0_{\pm7.2}}$ | 14.0 ↑ |
| diff | $\mathbf{100.0_{\pm0.0}}$ | $\mathbf{100.0_{\pm0.0}}$ | 0.0 ○ |
| larger_animal | $\mathbf{88.0_{\pm0.0}}$ | $59.3_{\pm0.9}$ | -28.7 ↓ |
| negation | $\mathbf{89.0_{\pm0.6}}$ | $88.0_{\pm0.6}$ | -1.0 ↓ |
| object_counting | $52.0_{\pm2.1}$ | $\mathbf{57.3_{\pm3.5}}$ | 5.3 ↑ |
| orthography_starts_with | $69.3_{\pm0.7}$ | $\mathbf{72.7_{\pm1.2}}$ | 3.3 ↑ |
| rhymes | $\mathbf{97.7_{\pm1.3}}$ | $68.0_{\pm8.5}$ | -29.7 ↓ |
| second_word_letter | $48.3_{\pm1.4}$ | $\mathbf{100.0_{\pm0.0}}$ | 51.7 ↑ |
| sentence_similarity | $\mathbf{32.7_{\pm1.2}}$ | $32.7_{\pm2.7}$ | 0.0 ○ |
| sentiment | $89.0_{\pm0.0}$ | $\mathbf{93.7_{\pm0.3}}$ | 4.7 ↑ |
| sum | $\mathbf{100.0_{\pm0.0}}$ | $\mathbf{100.0_{\pm0.0}}$ | 0.0 ○ |
| synonyms | $13.7_{\pm0.3}$ | $\mathbf{14.0_{\pm1.5}}$ | 0.3 ↑ |
| taxonomy_animal | $\mathbf{78.7_{\pm1.9}}$ | $78.3_{\pm6.8}$ | -0.3 ↓ |
| translation_en-de | $83.3_{\pm0.3}$ | $\mathbf{84.7_{\pm0.3}}$ | 1.3 ↑ |
| translation_en-es | $84.7_{\pm0.9}$ | $\mathbf{89.3_{\pm0.3}}$ | 4.7 ↑ |
| translation_en-fr | $85.0_{\pm1.0}$ | $\mathbf{88.0_{\pm0.6}}$ | 3.0 ↑ |
| word_sorting | $69.7_{\pm0.3}$ | $\mathbf{73.3_{\pm0.7}}$ | 3.7 ↑ |
| word_unscrambling | $\mathbf{62.0_{\pm2.1}}$ | $\mathbf{62.0_{\pm1.5}}$ | 0.0 ○ |
| LR (10% noise) | $\mathbf{42.3_{\pm0.9}}$ | $24.0_{\pm14.1}$ | -18.3 ↓ |
| LP-variant (10% noise) | $48.0_{\pm2.5}$ | $\mathbf{55.0_{\pm1.0}}$ | 7.0 ↑ |
| AG News Remap (10% noise) | $30.3_{\pm4.4}$ | $\mathbf{51.3_{\pm1.2}}$ | 21.0 ↑ |
| SST5 Reverse (10% noise) | $31.3_{\pm2.6}$ | $\mathbf{34.0_{\pm1.0}}$ | 2.7 ↑ |

Table 27: Test accuracies counterpart of Tab. 4 over 3 independent trials.

| AG News Remap (10% noise) | | | | SST5 Reverse (10% noise) | | |
|---|---|---|---|---|---|---|
| Size $n$ | EASE | EASE with retrieval | | Size $n$ | EASE | EASE with retrieval |
| 1000 | $35.3_{\pm5.8}$ | $\mathbf{40.7_{\pm0.9}}$ | | 1000 | $33.0_{\pm2.9}$ | $\mathbf{35.3_{\pm2.6}}$ |
| 10000 | $\mathbf{39.3_{\pm3.4}}$ | $31.3_{\pm5.5}$ | | 3000 | $\mathbf{39.0_{\pm1.1}}$ | $31.7_{\pm2.6}$ |
| 50000 | $32.0_{\pm1.0}$ | $\mathbf{36.0_{\pm1.0}}$ | | 5000 | $36.7_{\pm0.9}$ | $\mathbf{37.0_{\pm0.6}}$ |
| 100000 | $29.0_{\pm2.5}$ | $\mathbf{37.3_{\pm2.4}}$ | | 7000 | $23.0_{\pm2.1}$ | $\mathbf{37.3_{\pm0.9}}$ |

