# OpenReview forum: "Prompt Optimization with EASE? Efficient Ordering-aware Automated Selection of Exemplars"
_ICML.cc/2024/Workshop/ICL — ICML 2024 Workshop ICL Poster_

### Official Review · Reviewer_fbPV · 2024-06-07

**Rating:** 2
**Fit:** 3
**Confidence:** 2

**Workshop Review:**

Overall, I think the paper is solid, with clearly written context, a novel algorithm, and comprehensive evaluations to support its arguments.

I do have some questions, suggestions, and concerns that may help improve the draft:

1. Out-of-distribution validations and test set: The method relies on the validation data to select good exemplars. What will happen if the validation data and test data are not from the same distribution?

2. Use of validation data as in-context exemplars: Is it possible to use the validation data as the in-context exemplars, especially when the training set is noisy?

3. Better complexity analysis: Although there is a speedup analysis in the appendix, I still don't know how much time the algorithm will take.

**Reason For Not Giving Higher Score:**

There are some concerns about how the methods generalize to real-world applications.

**Reason For Not Giving Lower Score:**

It makes a valuable contribution to the ICL community.

---

### Official Review · Reviewer_NR88 · 2024-06-11

**Rating:** 2
**Fit:** 3
**Confidence:** 2

**Workshop Review:**

This paper explores the in-context learning setting of LLM, particularly, the ordering of the examples. It proposes a method called EASE based on the latent embeddings to find an ordering of the demonstrations that achieve good performance on 17 out of 19 tasks.

**Reason For Not Giving Higher Score:**

None

**Reason For Not Giving Lower Score:**

1.	Experiments include comprehensive baselines.
2.	The analysis is interesting.

---

### Meta-Review · Area_Chair_A6jt · 2024-06-14

**Recommendation:** 2

**Metareview:**

The present paper presents a method to improve performance of exemplar selection for ICL using a bandit style algorithm which accounts also for the ordering of the examples which tends to have a large impact on the ICL performance of LLMs.

Both reviewers find the submission a good topical fit for the workshop and find the algorithm "novel" and achieving good performance on 17 out of 19 tasks.

I recommend acceptance as a poster.

---

### Decision · Program_Chairs · 2024-06-17

Accept (Poster)